# CD20 tails interact with the 14-3-3/GEF-H1 complex and microtubule network upon PKCδ phosphorylation

Kathrin Kläsener [ID] [1,2,3,10 ✉], Cindy Eunhee Lee [ID] [4,5,10], Julian Bender [ID] [6], Angela Naumann [ID] [2,7], Lena Reimann [ID] [1], Geoffroy Andrieux[8], Claudio Mussolino[5,9], Nadja Herrmann [ID] [2], Roland Nitschke [ID] [2,7], Reinhard E Voll [ID] [3], Bettina Warscheid[6], Klaus Warnatz[3,9] & Michael Reth [ID] [1,2,3 ✉]

## Abstract

CD20 is a four-helix transmembrane protein specifically expressed in B-cells that serves as a prominent target of therapeutic anti-CD20 antibodies. It is localized in a membrane nanocluster harboring the B-cell antigen receptor of IgD class (IgD-BCR), where it functions to maintain the resting state of naïve B-lymphocytes. How CD20 exerts this resting B-cell gatekeeper function is not yet known. Using Ramos and human peripheral blood B-cells, we show here that the serine/threonine kinase PKCδ constitutively phosphorylates serine residues in the two cytosolic tails of CD20. Phosphorylated CD20 becomes a binding target for 14-3-3 adaptor proteins, which link it to the RhoA GDP/GTP exchange factor GEF-H1 and the microtubule network, supporting the function of the IgD-BCR nanocluster. Binding of anti-CD20 antibodies results in microtubule dissociation and replacement of the GEF-H1/CD20 complex with a RhoA-GTP/ROCK1/CD20 complex, which promotes actomyosin contractility. Our study suggests that CD20 not only maintains the resting state of B-lymphocytes by anchoring the microtubule network and controlling the stability of the IgD-BCR nanocluster, but also mediates the microtubule/actin switch in active B-lymphocytes. These findings could have important implications for anti-CD20 antibody treatment and the optimization of therapeutic protocols.

**Subject Categories** Cell Adhesion, Polarity & Cytoskeleton; Immunology

## Introduction

Mature naïve B cells co-express an IgD and an IgM-class B cell antigen receptor (IgD-BCR and IgM-BCR) that are localized in different membrane nanoclusters (Kläsener et al, 2014). The underlying cytoskeleton plays an important role in the location and stability of these nanoclusters (Mattila et al, 2013; Treanor et al, 2010). The IgD-BCR nanocluster has a raft-type lipid composition and harbors important B cell coreceptors, such as CD19, CD20, CD40, CXCR4, and the BAFF receptor (Becker et al, 2017; Hobeika et al, 2015; Maity et al, 2015). Interestingly, most proteins residing within the IgD nanocluster, including the IgD-BCR, reach their full expression level only at the mature naive B cell stage (Dirks et al, 2023). This is also the case for CD20, a 37 kD non-glycosylated phosphoprotein with four transmembrane domains that form dimers or higher oligomers on the B cell surface (Deans et al, 2002; Reinhardt et al, 2023; Rougé et al, 2020). The cDNA of human CD20 was first cloned in 1988 (Einfeld et al, 1988) and found to be encoded by the *MS4A1* gene that is one of the 18 members of the *MS4A* gene family. Due to the lack of a known ligand, the biological function of CD20 has been regarded as an enigma of B cell biology (Pavlasova and Mraz, 2020; Riley and Sliwkowski, 2000). However, newer data show that CD20 is a highly interactive protein that helps in the regulation and functional organization of other receptors on the B cell surface (Müller et al, 2021). For example, CD70 has recently been shown to require the presence of CD20 for immune synapse formation and efficient T:B collaboration (Arp et al, 2025).

CD20 is best known as a prominent target of therapeutic monoclonal antibodies (mAb) such as rituximab (RTX), obinutuzumab (GA101), or ocrelizumab used for the treatment of B cell chronic lymphocytic leukemia (B-CLL), malignant B cell lymphomas or human autoimmune diseases such as rheumatoid arthritis (RA), granulomatosis with polyangiitis (GPA), systemic lupus

[1]Biology III, Faculty of Biology, University of Freiburg, Freiburg, Germany. [2]Signaling Research Centers CIBSS and BIOSS, University of Freiburg, Freiburg, Germany. [3]Department of Rheumatology and Clinical Immunology, Medical Center –Faculty of Medicine, University of Freiburg, Freiburg, Germany. [4]Faculty of Science and Technology, University of Canberra, Canberra, Australia. [5]Institute for Transfusion Medicine and Gene Therapy, Medical Center - University of Freiburg, Freiburg, Germany. [6]Biochemistry II, Theodor-Boveri-Institute, Biocenter, Faculty of Chemistry and Pharmacy, University of Würzburg, Würzburg, Germany. [7]Life Imaging Center and Signalling Research Centres CIBSS and BIOSS, University of Freiburg, Freiburg, Germany. [8]Institute of Medical Bioinformatics and Systems Medicine, Medical Center - Faculty of Medicine, University of Freiburg, Freiburg, Germany. [9]Center for Chronic Immunodeficiency (CCI), Medical center - Faculty of medicine, University of Freiburg, Freiburg, Germany. [10]These authors contributed equally: Kathrin Kläsener, Cindy Eunhee Lee. ✉E-mail: kathrin.klaesener@bioss.uni-freiburg.de; michael.reth@bioss.uni-freiburg.de

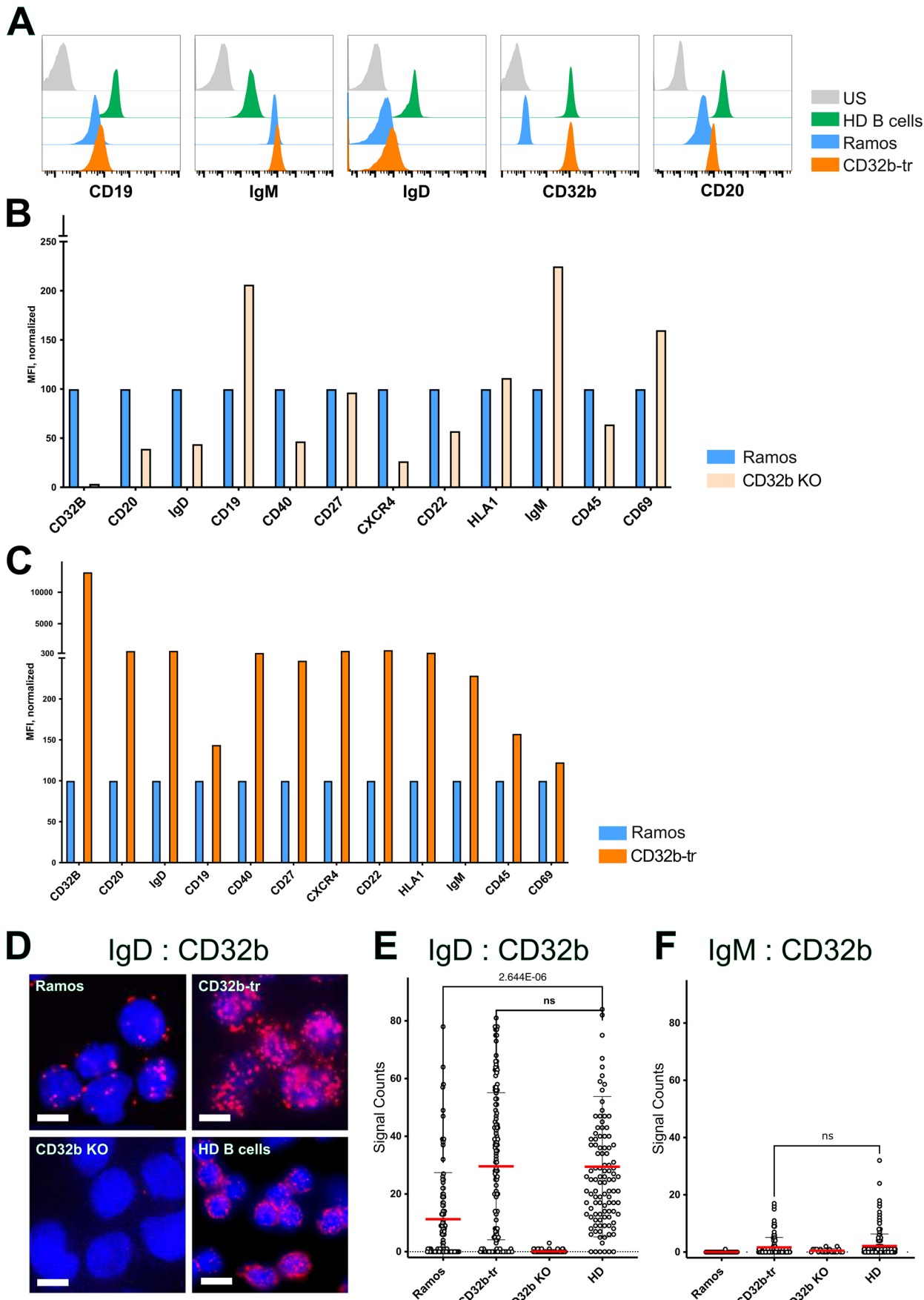

**Figure 1. Functional association of the Fc-receptor CD32b with the IgD-BCR nanocluster.**

(A) Expression of the FcγRIIb receptor on Ramos, changes the surface abundance of critical B cell proteins to that of naïve B cells. Flow cytometry analysis shows the comparison of B cell surface protein abundance on healthy donor (HD) negative selected naïve B cells (green), Ramos (blue) and FcγRIIb receptor-transfected (CD32b-tr, orange) Ramos cells. Unstained (US) control in gray, n > 10. (B, C) Flow cytometry analysis of the surface abundance of B cell marker proteins of CD32b KO (beige) or CD32b-tr (orange) in comparison to WT Ramos B cells (blue). Expression data of CD32b KO or CD32b-tr cells are normalized to the WT Ramos set to 100%, n = 4. (D) Fab-PLA study of the IgD-BCR:CD32b proximity on the surface of WT Ramos (upper left), CD32b-tr (upper right), CD32b KO (lower left), and HD B cells (lower right). PLA dots in red and nuclei stained blue by DAPI, Scale bar 5 μm. Quantification of Fab-PLA data of (E) IgD:CD32b and (F) IgM:CD32b with CellProfiler, n = 3. Calculation of the significance between the groups with PRISM 10, one-way ANOVA. Exact p values were calculated with excel =T.DIST. Unpaired Student's t-test shows no significant difference between CD32-tr and HD B cells. Red bars show mean values. Source data are available online for this figure.

erythematosus (SLE) and, more recently, multiple sclerosis (MS) (Jazirehi and Bonavida, 2005; Klein et al, 2021; Kosmas et al, 2002; Margoni et al, 2022; Roubaud-Baudron et al, 2012; Smith, 2003). The anti-CD20 mAbs are thought to eliminate targeted B cells by antibody-dependent cellular cytotoxicity (ADCC) mechanisms, phagocytosis (Payandeh et al, 2019), or by induction of cell cycle arrest and initiation of an apoptosis program in certain B cell lines (Boross and Leusen, 2012). A more profound insight into the biological function of CD20 could improve the success of anti-CD20 immunotherapy, as relapses occur repeatedly after treatment (Marshalek et al, 2022; Rezvani and Maloney, 2011; Rushton et al, 2020). An important advance towards this goal is the finding that CD20 is part of the IgD-BCR nanocluster, where it functions as a gatekeeper for the resting state of naïve human B lymphocytes (Kläsener et al, 2021). Two other regulators of resting B cells are the inhibitory Fc-receptor FcγRIIb (CD32b) and the serine/threonine protein kinase C delta (PKCδ), a member of the novel serine/threonine protein kinase C (nPKC) (Miyamoto, 2002; Nowicka et al, 2021; Salzer et al, 2016; Vaughan et al, 2014). Here, we show that CD32b and PKCδ support the gatekeeper function of CD20 within the IgD-BCR nanocluster. While CD32b increases the IgD-BCR/CD20 expression, PKCδ phosphorylates serine residues in the N- and C-terminal cytosolic tails of CD20. In this way, PKCδ couples CD20 to the microtubule (MT) network in resting and the RhoA/ROCK1 signaling pathway in RTX-exposed B cells.

## Results

### CD32b is associated with the IgD-BCR nanocluster and stabilizes CD20 expression

The Burkitt lymphoma cell line Ramos is a useful and well-characterized model for studying human B cell functions using modern engineering tools, such as the CRISPR/Cas9 technique and transgenesis (Adli, 2018; Webster et al, 2019). Flow cytometry analysis shows that Ramos express more IgM-BCR and less IgD-BCR than human healthy donor (HD) B cells (Fig. 1A). The expression of the B cell coreceptor CD19 is comparable on the surfaces of the two B cells, whereas the inhibitory Fc-receptor CD32b and CD20 are less well expressed on Ramos B cells. Transfection of Ramos B cells with a CD32b lentiviral expression vector resulted in the isolation of a Ramos CD32b-tr B cell line displaying CD32b expression similar to that of HD B cells. Compared to the untransfected (Ramos), the CD32b-tr B cells express more CD20 and IgD-BCR on their cell surface. CD32b-tr B cells also display a higher abundance of other IgD-BCR nanocluster

components, such as CD40 and CXCR4 (Fig. 1B). To analyze the impact of CD32b on B cell surface marker expression, we generated CD32b-deficient (CD32b KO) Ramos clones using the CRISPR/Cas9 technique. Loss of CD32b resulted in a reduced expression of proteins associated with the IgD-class nanocluster such as CD20, CD40, CXCR4, and the IgD-BCR, and an increased expression of CD19, IgM-BCR, and the B cell activation marker CD69 (Fig. 1C). Notably, CD19 and the IgM-BCR interact with each other on activated B lymphocytes (Depoil et al, 2008; McCaleb et al, 2023) and the expression pattern of CD32b KO Ramos cells suggests that the loss of CD32b results in a partial B cell activation. The effect of CD32b on the expression of B cell surface markers suggests that CD32b is part of the IgD-BCR nanocluster. To test for this, we employed the Fab-based proximity ligation assay (Fab-PLA), which monitors the proximity between CD32b and the IgD-BCR within a range of 10–20 nm on the cell surface (Fig. 1D,E) (Kläsener et al, 2018). Compared to Ramos, the CD32b:IgD PLA signal is increased in Ramos CD32-tr cells and is similar to that of HD B cells. As a negative control for this analysis, we tested Ramos CD32b KO cells, which showed no PLA signal. A parallel CD32b:IgM Fab-PLA revealed minimal interaction between CD32b and the IgM-BCR on the human B cells studied (Fig. 1F). Taken together, this analysis shows that CD32b is part of the IgD-BCR nanocluster, where it acts as a gatekeeper for the resting state of human B cells. This conclusion is in line with the finding that CD32b mutations are associated with human autoimmune diseases such as systemic lupus erythematosus (SLE) (Sjef Verbeek et al, 2019).

### CD20 is a substrate of PKCδ and acts as a gatekeeper for the resting state of B cells

PKCδ has been found to be constitutively associated with the BCR signaling subunit Igα (CD79a) and to negatively regulate BCR signaling (Barbazuk and Gold, 1999; Pracht et al, 2007; Salzer et al, 2016). To study the gatekeeper function of PKCδ, we employed the CRISPR/Cas9 method to generate two PKCδ-deficient (PKCδ KO) clones of the human Burkitt lymphoma cell line Ramos (Appendix Fig. S1A,B). The PKCδ KO Ramos cells were viable and could be maintained in culture. First, we compared the expression of B-cell marker proteins on the surface of the PKCδ KO line and the Ramos cells using flow cytometric analysis (Fig. 2A). Normalization of the protein expression levels on PKCδ KO to Ramos cells shows that the loss of PKCδ is associated with a 25-fold increase in the expression of the B cell activation marker CD69 on Ramos B cells. The PKCδ deficiency also results in the upregulation of membrane proteins that are associated with the IgD-class nanocluster on resting B cells, including CD20, CD19, CD40, and the IgD-BCR,

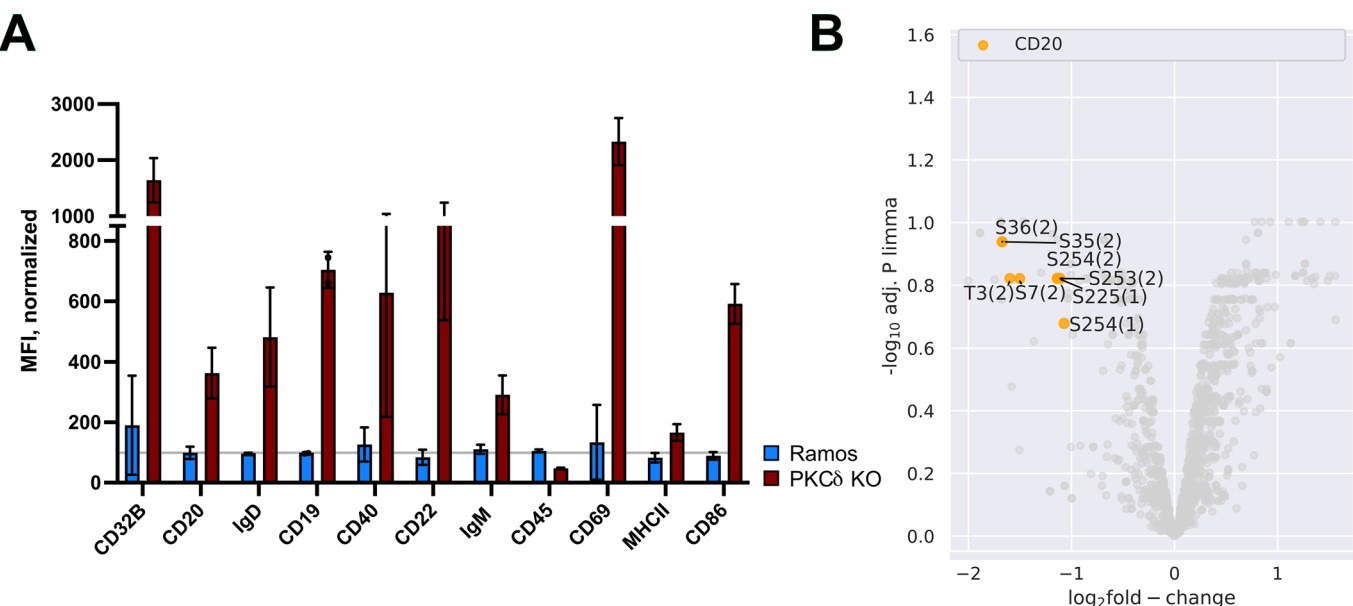

**Figure 2. Phenotype of PKCδ KO Ramos B cells and identification of PKCδ substrate proteins.**

(A) Comparison of the expression of B cell marker protein on the surface of PKCδ KO (brown) and WT Ramos B cells (blue) by flow cytometry. The surface marker detection by flow cytometry of the two PKCδ KO Ramos B cell clones was determined at least six times in different time periods of cells in cell culture, $n = 6$. The expression data of PKCδ KO are normalized to WT Ramos cells, set to 100%, shown is the mean with SD. (B) Volcano plot showing quantitative changes of constitutive serine/threonine phosphorylation of substrate proteins of WT in comparison to PKCδ KO Ramos B cells. Plot X-axis depicts log2 fold change of phosphorylated serine or threonine residues in the whole proteome of the two cell lines. The Y-axis shows the log10 FDR-adjusted $p$ value <0.05. The significant changes of phosphorylated MS4A1 serine residues are indicated in orange (multiplicities in brackets). The analysis of the whole phosphoproteome was performed with one PKCδ KO B cell clone in four replicates. Source data are available online for this figure.

whereas proteins that are associated with the IgM-BCR, such as CD45, are downregulated. The 20-fold upregulation of CD32b on PKCδ KO Ramos B cells suggests that the gatekeeper function of CD32b compensates for the loss of PKCδ.

Next, we conducted a phosphoproteome analysis to monitor differences in constitutive serine/threonine phosphorylation levels in Ramos and the PKCδ KO cell lines (Fig. 2B; Dataset EV1). This analysis identified several serine residues in the N- and C-terminal cytosolic tails of CD20 that are constitutively phosphorylated by PKCδ. Thus, PKCδ and CD20, two gatekeepers, which are both involved in regulating the resting state of B lymphocytes, are connected by an enzyme/substrate relationship. Global serine/threonine phosphoproteome analysis of Ramos and PKCδ KO cells also revealed specific alterations in the phosphorylation of components of the RhoA/Rock-1 pathway and the MT cytoskeletal network, as described below (Appendix Fig. S1C). Comparing the RNAseq data from the two cell lines revealed that PKCδ KO cells displayed increased expression of genes encoding for proteins involved in MT depolymerization and loss of cortical cytoskeletal stability compared to Ramos cells (Appendix Fig. S2A,B, key genes listed in Appendix Fig. S2C; Dataset EV1).

## The function of CD20 serine residues and their association with 14-3-3 adapter proteins

Analysis of the serine/threonine phosphoproteome of Ramos cells and their PKCδ-deficient subclones suggests that four serine residues in the N-terminal cytosolic tail (S7, S35/36, and S49)

and three serine residues in the C-terminal cytosolic tail (S225 and S253/254) of CD20 are constitutively phosphorylated by PKCδ. Interestingly, the three most significant CD20 serine phosphorylation sites (S35/36 and S225) are found within an RxxS motif (Gogl et al, 2021). Once phosphorylated, this motif could serve as a binding target for the 14-3-3 adapter family proteins, which function by binding to serine/threonine-phosphorylated intracellular proteins, thereby altering their binding partners' conformation, activity, and subcellular localization (Fig. 3A) (Johnson et al, 2010; Obsilova and Obsil, 2022; Pitasse-Santos et al, 2024; Xiao et al, 1995). Additionally, another serine residue (S221) is located near the plasma membrane. Neither the phosphorylated nor the unphosphorylated form of S221 was detected in the phosphoproteome, possibly due to adjacent palmitoylation at C220 (Ivaldi et al, 2012). S221 lies within a classical RxxSxP binding motif for 14-3-3 proteins. To test for the function of the identified serine residues, we generated expression vectors for WT or mutant CD20 proteins carrying a serine to alanine (S/A) exchange of either the four N-terminal (Nmut), the four C-terminal (Cmut), or all eight serine residues (NCmut). To eliminate the possibility of 14-3-3 binding entirely, S221 was also mutated to alanine. For purification purposes, the expressed CD20 proteins also carry a C-terminal flag-tag (Fig. 3B). Next, we used the CRISPR/Cas9 method to generate CD20-deficient Ramos B cells that were then transfected with retroviral expression vectors encoding either the WT CD20 or tail mutants (Nmut, Cmut, or NCmut) of CD20. The obtained CD20 transfectants were sorted and then transfected again with the human CD32b lentiviral expression vector. Flow cytometry analysis

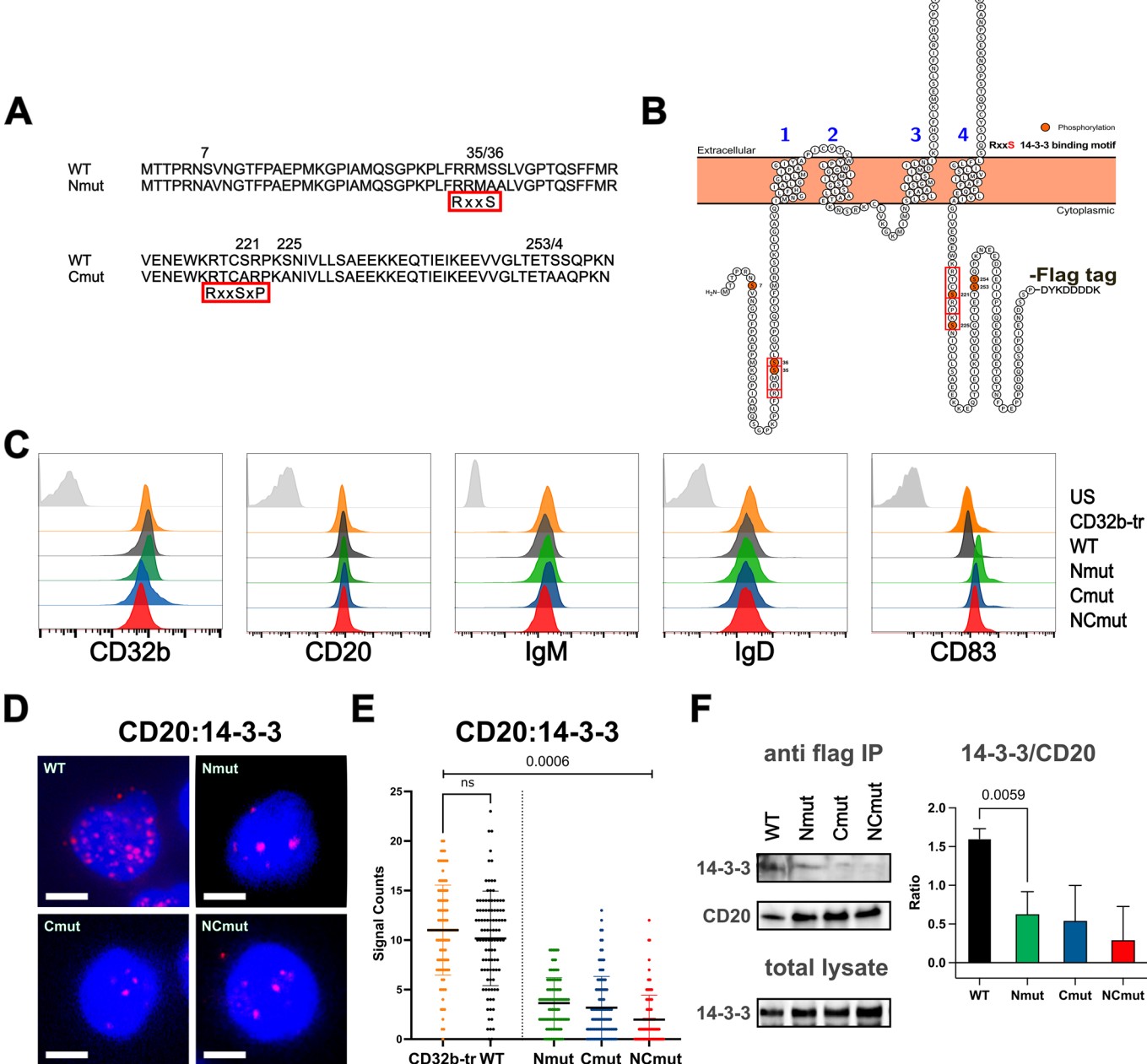

Figure 3. Mutational analysis of CD20 serine residues and their association with 14-3-3.

(A) Protein sequence of the N-terminal (top) and part of the C-terminal cytoplasmic tail of human CD20 (bottom). Below the WT sequence, the sequences with the introduced S/A mutations (bold letters) at the N-terminal (Nmut) and C-terminal (Cmut) cytoplasmic tails. The location of the RxxS motifs implicated in 14-3-3 binding is shown in red squares. (B) Membrane topology of human CD20 with a C-terminal flag-tag (CD20-flag). Red boxes indicate the location of the RxxS motifs. (C) Flow cytometry analysis of CD32b-tr Ramos cells (orange) and reconstituted CD20 WT Ramos cells (black), Nmut (green), Cmut (dark blue) or NCmut (red). Expression of CD20, CD32b, IgM, IgD, and CD83 is presented. Unstained (US) CD32b-tr cells in gray, $n = 6$. (D) Fab-PLA study of the CD20:14-3-3 proximity in WT (upper left), Nmut (upper right), Cmut (lower left), and NCmut (lower right) Ramos B cells. PLA dots in red and nuclei stained blue by DAPI, Scale bar 5 µm. (E) Quantification of PLA data by CellProfiler. Significant differences between groups of mutated CD20 cells to unmutated Ramos or HD B cells were calculated with PRISM10, one-way ANOVA. Unpaired Student's $t$-test shows no significant difference between the mean values of CD32-tr and WT Ramos cells. $T$-test and ANOVA after PLA experiments were performed for at least three independent PLA studies of the different batches of CD20 mutants, $n = 3$. (F) Western blot analysis of the CD20-flag/14-3-3 association in WT, Nmut, Cmut, and NCmut Ramos B cells. The immunoprecipitates of 14-3-3 protein in anti-flag and total lysates is shown on the left, on the right, the ratio of 14-3-3 to CD20 is quantified with Image Studio Lite (Licor), showing the difference in signal intensity in the immunoprecipitates calculated by unpaired $t$-test; mean with SD above. Lysates were taken from at least three independently generated Ramos cell lines. Source data are available online for this figure.

showed that all four double-transfectants expressed CD20 and CD32b at levels comparable to those of CD32b-tr Ramos line (Fig. 3C). Similar expression was also found for CD19, as well as for the IgM- and IgD-BCR. However, in comparison to the CD32b-tr Ramos line and the CD20 WT transfectants, Ramos cells expressing the Nmut, Cmut, or NCmut CD20 mutants displayed a higher expression of the activation marker CD83, suggesting that the S/A exchange partially compromises the gatekeeper function of CD20 (Fig. 3C, far right) (Z. Li et al, 2019).

The finding that four of the eight serine residues under study are part of an RxxS motif sequence suggests that CD20 is not only constitutively phosphorylated by PKCδ but also constitutively associated with a 14-3-3 adapter protein. To test for this, we used the 1-PLA assay to monitor the colocalization of CD20 with a 14-3-3 adapter protein within CD32b-tr Ramos cells and CD32b/CD20 double-transfectants that express, in addition to CD32b, either CD20 WT or one of three different tail mutants of CD20 (Fig. 3D,E). The 1-PLA assay employs CD20 and 14-3-3-specific DNA-coupled antibodies that detect a colocalization of the two proteins in the 10-40 nanometer range. Colocalization of CD20 with 14-3-3 was detected in the CD20 WT cells, but to a lesser extent in the CD20Nmut, Cmut, and NCmut Ramos cells. Western blot analysis of anti-flag immunoprecipitates of Ramos cell lysates confirms that the S/A mutants of CD20 are less well associated with the 29 kD 14-3-3 protein compared to CD20 WT (Fig. 3F).

## The GEF-H1 activity is regulated by the cytoplasmic tails of CD20

The 14-3-3 protein is a dimeric adapter that can stabilize CD20 homodimers and/or heterodimeric associations with other RxxS motif-containing proteins (Paul et al, 2012). In a preliminary study, we found that exposing B cells to anti-CD20 antibodies activates the RhoA/Rock1 signaling pathway (see below) (Ricker et al, 2016). A key activator of this pathway is the GDP/GTP exchange protein GEF-H1 (also known as ARHGEF2), which is constitutively associated with the MT network in resting B cells (Birkenfeld et al, 2008; Krendel et al, 2002). GEF-H1's C-terminal tail contains intrinsically disordered regions that carry an RRxxR and an RxxSxP motif (Fig. 4A). The RRxxR motif is a known binding target for the light chains of the dynein motor complex (such as LC8 or Tctex-1) and is thought to be one of the interfaces that link GEF-H1 to MTs (Joo and Olson, 2021; Meiri et al, 2014). The serine residue S886 within the RxxSxP motif of GEF-H1 is phosphorylated by the p21-activated kinase 2 (PAK2) (Zenke et al, 2004). This modification inhibits GEF-H1 activity. The binding of a 14-3-3 adapter protein to the phosphorylated RxxSxP motif may connect GEF-H1 to the constitutively phosphorylated CD20 tails. We used the 1-PLA assay to test for this and found a colocalization of CD20 and GEF-H1 in the CD20 WT transfectant and, to a lesser extent, in the CD20Nmut, Cmut or NCmut transfected Ramos cells (Fig. 4B,C). Western blot analysis of anti-Flag immunoprecipitates of CD20 proteins in the Ramos cell lysates confirmed that the tail mutants of CD20 are less well associated with GEF-H1 than CD20 WT (Fig. 4D). In particular, the NC double mutant of CD20 is barely associated with GEF-H1, compared to CD20 WT.

The amount of active GEF-H1 protein involved in the GDP/GTP exchange in the cytosol of Ramos B cells can be tested using a pull-down assay that employs a G17A trapping mutant of RhoA,

which stabilizes the RhoA/GEF-H1 intermediate interaction (Waheed et al, 2012). For this, lysates of the different Ramos cell lines were incubated with GST-RhoA-G17A-coupled GST beads, and the bound GEF-H1 was monitored by Western blotting. Compared to CD20 WT, the cytosol of Ramos cells expressing the Nmut, Cmut, or NCmut contained higher amounts of active GEF-H1/RhoA complexes (Fig. 4E,F). Taken together, these data suggest that the constitutively phosphorylated serine residues within the cytosolic tails of CD20 regulate the RhoA/Rock-1 pathway by binding and sequestering GEF-H1.

## Exposure to the anti-CD20 antibody activates the RhoA/Rock-1 pathway

We next asked whether or not the exposure of Ramos B cells to a therapeutic anti-CD20 antibody, such as rituximab (RTX), alters the interaction between CD20 and components of the RhoA/Rock-1 pathway. For this, we employed the 1-PLA assays to monitor changes in the proximity of CD20 to 14-3-3, GEF-H1, RhoA, and Rock-1 in Ramos CD32b-tr as well as in CD20 WT or CD20 NCmut transfectants before and after a 5 min exposure to RTX (Fig. 5). The RTX treatment reduced the CD20/14-3-3 proximity in CD32b-tr and CD20 WT-transfected Ramos cells to the level of unstimulated NCmut Ramos cells (Fig. 5A,B) as well as the CD20/GEF-H1 proximity (Fig. 5C,D). Interestingly, the 1-PLA assays revealed an increased proximity between CD20 and RhoA (Fig. 5E,F) as well as between CD20 and Rock-1 in RTX-treated Ramos cells expressing CD20 WT, but not in cells expressing CD20 NCmut. These results suggest that in RTX-treated B cells, CD20 stabilizes a RhoA/Rock-1 complex at the inner leaflet of the plasma membrane in a 14-3-3-dependent manner. Importantly, the C-terminus of Rock-1 also carries an evolutionarily conserved RxxS motif (Chuang et al, 2013). Thus, it is feasible that CD20 functions as both a gatekeeper of the resting state, and a switch that promotes activation of the RhoA/Rock-1 pathway in B cells exposed to antigen or RTX. As a control, we examined the formation of active GEF-H1 after RTX activation and GDP/GTP exchange. The pull-down assay with the G17A trapping mutant of RhoA showed an increase in active GEF-H1/RhoA complexes, especially in Ramos, CD32b-tr, and WT cells (Appendix Fig. S3A,B).

We then used the phosphoflow assays to monitor alterations in the phosphorylation of the RhoA/Rock-1 pathway members after RTX treatment of CD32b-tr, CD20 WT or NCmut Ramos cells (Appendix Fig. S4). Exposure of CD32b-tr or CD20 WT Ramos cells to RTX resulted in a reduction of the S886 phosphorylation of GEF-H1 and the loss of the inhibitory S188 phosphorylation of RhoA (Appendix Fig. S4A,B) (Birkenfeld et al, 2008; Patel and Karginov, 2014). The same treatment increased the phosphorylation of Rock1 tyrosine 914 (Y914), a modification that is associated with Rock-1 activation (Appendix Fig. S4C). Importantly, all these amino acids are phosphorylated less efficiently in the NCmut Ramos cells, and their phosphorylation state did not significantly change after RTX exposure. Together, these data demonstrate that the studied cytosolic serine/threonine residues connect CD20 to the RhoA/Rock-1 signaling pathway.

Exposure to the therapeutic antibody RTX activates the RhoA/Rock-1 signaling pathway not only in Ramos but also in naïve human B cells. This is indicated by the time-dependent alteration of the phosphorylation status of GEF-H1, RhoA and Rock-1

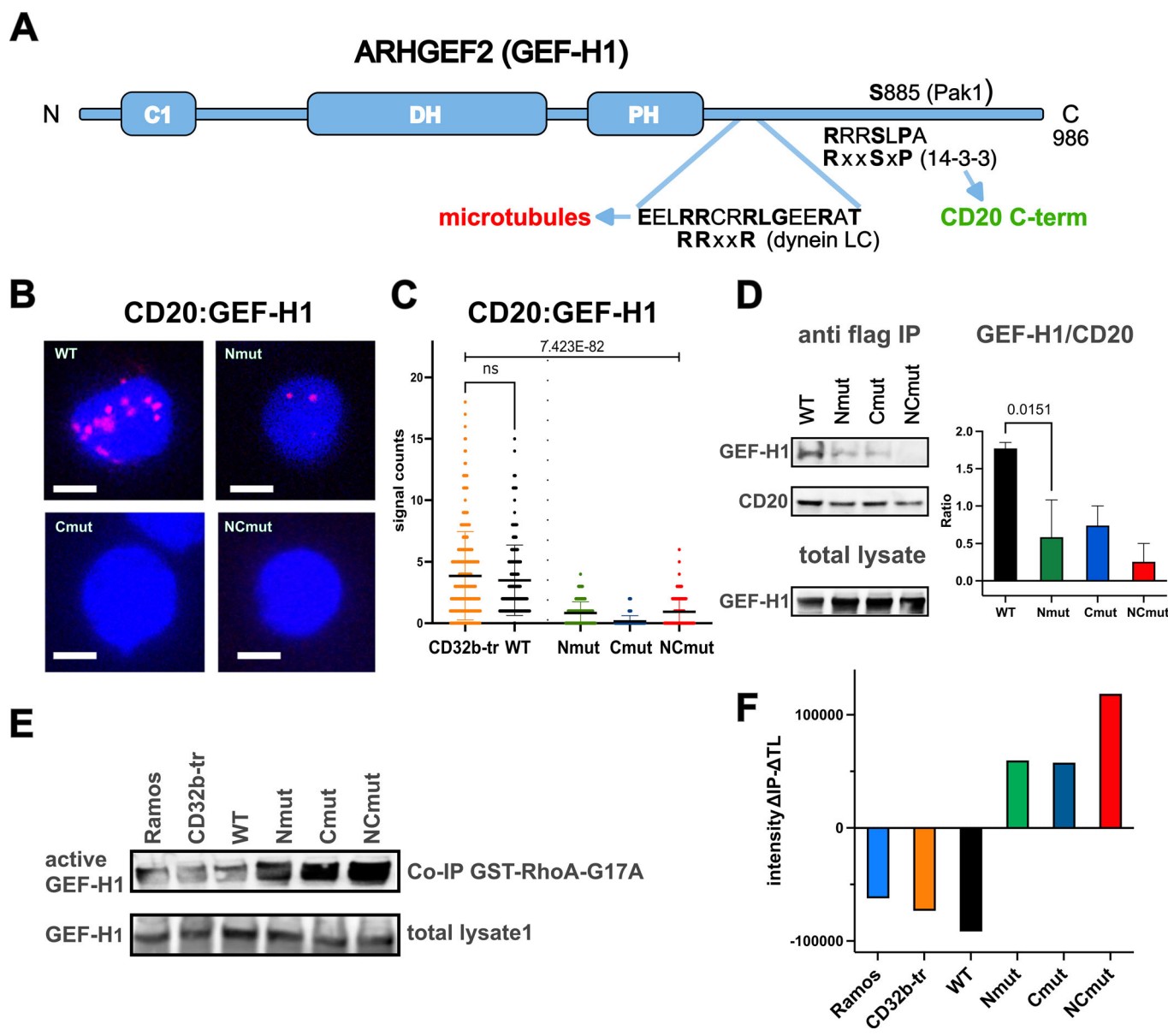

**Figure 4. Constitutive association of CD20 with the GDP/GTP exchange factor GEF-H1.**

(A) Schematic drawing of the domain structure of the RhoA guanine nucleotide exchange factor GEF-H1 (ARHGEF2). The functional domains are drawn as blue boxes. The location of the RRxxR motif implicated in dynein light chain binding (LC), and the Pak2 phosphorylated S886 within the 14-3-3 binding RxxSxP motif is indicated. (B) 1-PLA study of the CD20:GEF-H1 proximity in WT (upper left), Nmut (upper right), Cmut (lower left), and NCmut (lower right) Ramos B cells. PLA dots in red and nuclei stained blue by DAPI, Scale bar 5 μm, $n = 3$. (C) Quantification of PLA of B. with CellProfiler. Significant differences between groups were calculated using PRISM 10, one-way ANOVA. Exact $p$ values were calculated with excel =F.DIST.RT(F, DFN, DFD). Unpaired Student's $t$-test shows no significant difference between mean values of CD32-tr and WT Ramos cells, mean values with SD are presented, $n = 3$. (D) Western blot analysis of the constitutive CD20-flag/GEF-H1 association in WT, Nmut, Cmut, and NCmut Ramos B cells. The immunoprecipitates of GEF-H1 protein in the anti-flag and total lysates are shown on the left, and the respective calculations on the right. Statistical analysis of the difference in signal intensity (in pixels) measured was calculated by an unpaired $t$-test, mean values with SD are presented, Western blot analysis and CD20-flag/GEF-H1 association was performed on three independently obtained immunoprecipitates of CD20 mutants, $n = 3$. (E) Western blot analysis of free GEF-H1 in the lysates of Ramos and CD32b-tr cells, as well as reconstituted Ramos cells expressing CD20 WT, Nmut, Cmut or NCmut. Amount of GEF-H1 purified with GST-RhoAG17A beads and in total lysate is shown at the top and bottom, respectively. (F) Quantification of the Western blot data with Image Studio Lite (Licor) showing the difference in signal intensity (in pixels) in the immunoprecipitates (IP) versus total lysates (TL). Source data are available online for this figure.

when HD B cells are treated with 10 ug/mL of RTX (Appendix Fig. S4D–F).

The 1-PLA assays of HD B cells before or after a 5 min exposure to RTX showed a reduction in the proximity of 14-3-3 and GEF-H1 to CD20 (Fig. 6A–D, respectively) and at the same time a strong recruitment

of RhoA and Rock-1 in the vicinity of CD20 in the RTX-treated cells (Fig. 6E–H, respectively). Interestingly, exposure of HD B cells to an MT polymerization inhibitor, such as nocodazole (Noc), increased the association of CD20 with the RhoA/Rock-1 complex, similar to the effect of RTX treatment (Fig. 6E–H). Thus, the CD20 engagement not

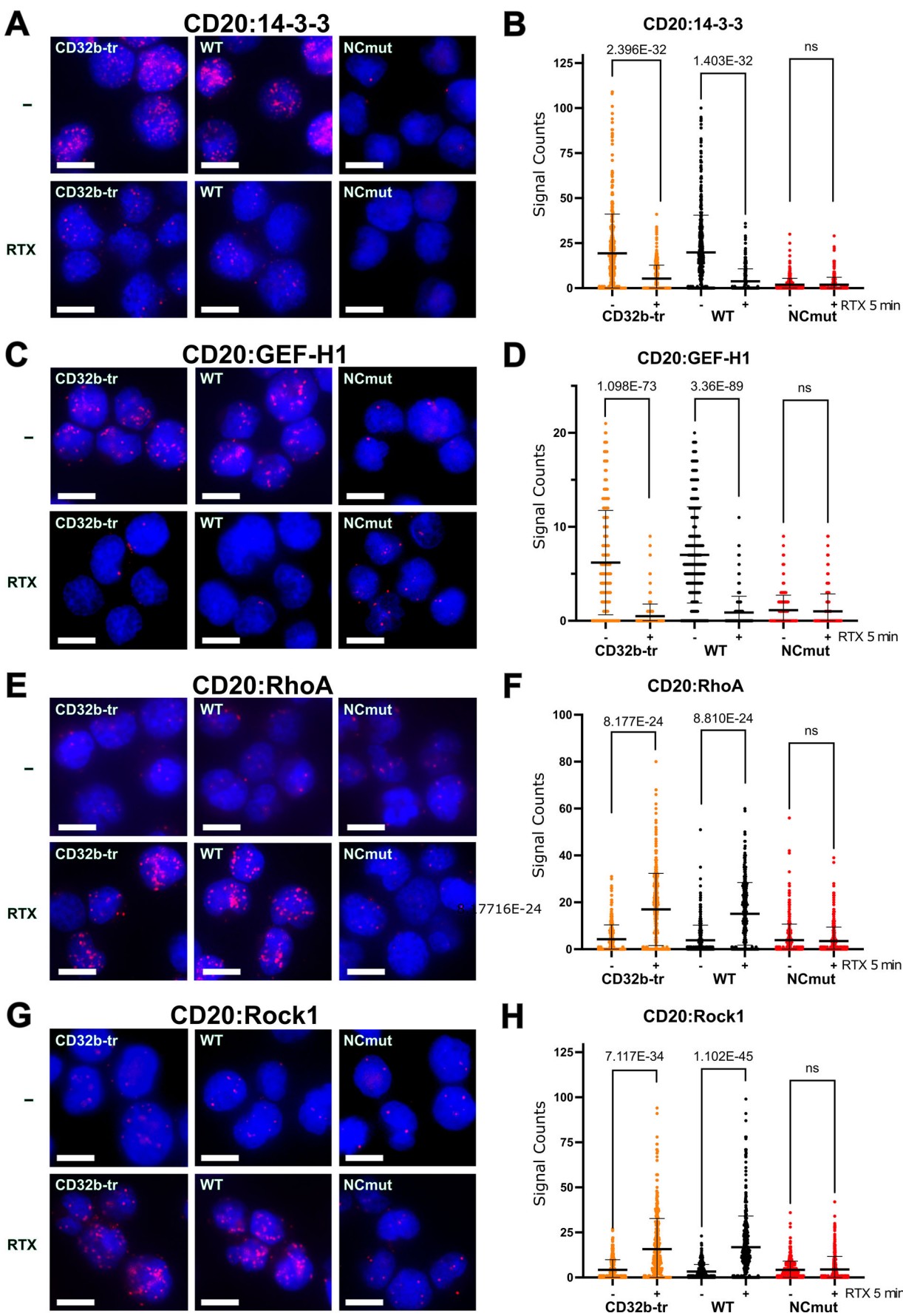

Figure 5.   Loss of GEF-H and gain of the RhoA/Rock1 association of CD20 upon RTX treatment.

PLA studies of Ramos cell proximities in untreated (top) or 5-min RTX-exposed (bottom) CD32b-TR Ramos cells (left), as well as in reconstituted Ramos cells expressing WT (middle) or NCmut CD20 (right), with corresponding quantification using CellProfiler. (A) PLA study of CD20:14-3-3, (B) Quantification of (A); (C) PLA Study of CD20:GEF-H1, (D). Quantification of (C); (E) PLA study of CD20:RhoA (F) Quantification of (E); (G) PLA study of the CD20:ROCK1 proximity. (H) Quantification of (G); all PLA experiments were performed at least $n = 3$. PLA dots in red and nuclei stained in blue by DAPI. Scale bar 10 µm. PRISM10 was used to calculate significant differences in mean values quantification between the two groups using an unpaired Student's $t$-test, all exact $p$ values were calculated with excel =T.DIST. Source data are available online for this figure.

only liberates and activates GEF-H1 but also results in an increased coupling of the engaged CD20 to the RhoA/Rock-1 signaling module.

## Anti-CD20 antibodies destabilize MTs and the IgD-BCR nanocluster

GEF-H1 is well-known to be associated with the MT network, and MT destabilization results in GEF-H1 liberation and increased activity (Azoitei et al, 2019; Krendel et al, 2002; Pineau et al, 2022). The result of our 1-PLA assay (see above) suggests that RTX activates the RhoA/Rock-1 signaling pathway by destabilizing the MT cytoskeleton. To test for this, we pretreated HD B cells for one hour with either Noc or with the MT stabilizer docetaxel (Tax) (Chang et al, 2008; Si et al, 2003; Steinmetz and Prota, 2018), then measured RTX-induced changes using flow cytometry (Fig. 7A,B). RTX-induced reductions in CD20 abundance and increased Rock1 phosphorylation can be prevented by pretreating the HD B cells with Tax for one hour. However, Noc-pretreated HD B cells showed already Rock1 activation with no further increase upon RTX treatment. Thus, the immediate effect of RTX binding to CD20 appears to be MT network destabilization, resulting in GEF-H1 liberation and the activation of the RhoA/Rock-1 signaling pathway.

MT dynamics are tightly regulated (Downing and Nogales, 2010; Janke and Magiera, 2020; Mandelkow et al, 1991). To test whether RTX binding to CD20 can influence or destabilize MTs, Ramos CD32b-tr cells were treated with RTX and labeled with an anti-alpha-tubulin antibody to visualize tubulin structures by immuno-fluorescence microscopy. The Airyscan superresolution imaging (Appendix Fig. S5) revealed the peeling of MT polymers, and the outwardly curved spiral protofilaments at the ends of shortening MT upon RTX exposure (Appendix Fig. S5A,B) (Downing and Nogales, 2010). In a second approach with HD naïve B cells (Appendix Fig. S6A–C), Airyscan confocal images of the labeled α-tubulin showed a spatial distribution of small "ram's horn" MT intermediates upon RTX treatment (Appendix Fig. S6B) (Gudimchuk and McIntosh, 2021; Hancock, 2015). Interestingly, pretreating with Tax before RTX exposure could stabilize the B cell integrity and reduce MT depolymerization (Appendix Fig. S6C).

In resting B cells, CD20 and CD19 reside in the same protein/lipid nanocluster in close proximity to the IgD-BCR, and the exposure to RTX leads to IgD-BCR/CD19 dissociation and the formation of an IgM-BCR/CD19 complex (Kläsener et al, 2021). Taking advantage of the Fab-PLA technique, we next asked whether the MT network is involved in the nanoscale receptor organization and reformation. Indeed, exposure of the HD B cells to either RTX or Noc dissolved the IgD-BCR nanocluster and resulted in IgM-BCR/CD19 conjugation. The RTX-induced nanoscale receptor rearrangements are partially prevented by pretreating HD B cells with Tax (Fig. 7C–F). Taken together, these data

demonstrate that, in resting B cells, the IgD-BCR nanocluster is connected to the MT network and that the cytosolic tails of CD20 play a crucial role in this connection and its formation.

## Discussion

### PKCδ and CD20: a kinase/substrate pair with a gatekeeper function

We have previously shown that CD20 is part of the IgD-BCR membrane nanocluster and acts as a gatekeeper for the resting state of B lymphocytes (Kläsener et al, 2021). Here, we describe that serine residues within the cytoplasmic tails of CD20 play an important role in this gatekeeper function. Specifically, we show that six serine residues of CD20 are constitutively phosphorylated by the serine/threonine kinase PKCδ, and that this modification couples CD20 to the MT network in resting B cells, and to the RhoA/ROCK1 signaling pathway in activated B cells. Exchanging the serine residues for alanine did not alter the amount of CD20 on the B cell surface, but it has an impact on the B cell activation state as CD20-mutated Ramos cells displayed higher expression of the activation marker CD83. These data suggest that the phosphorylated serine residues are connected to the gatekeeper function of CD20. Several genetic data support the notion that PKCδ also acts as a gatekeeper for the resting state of B lymphocytes (Barbazuk and Gold, 1999; Salzer et al, 2016). PKCδ KO mice develop autoimmunity, and loss-of-function PKCδ mutations in humans are associated with autoimmune diseases such as systemic lupus erythematosus (SLE) and B cell lymphoproliferative syndrome (S. Li et al, 2020; Sun Kuehn et al, 2013). Our finding that CD20 is a substrate of PKCδ provides novel mechanistic insights into the development of these autoimmune diseases.

### CD20 binds GEF-H1 and inhibits the GDP/GTP exchange activity in resting B cells

Four of the eight mutated serine residues of CD20 are part of conserved RxxS or RxxSxP motifs, which are well-known binding sites for 14-3-3 adapter proteins (Tinti et al, 2012). Constitutive CD20/14-3-3 association in resting human B cells was confirmed by 1-PLA and immunoprecipitation. The RxxS and RxxSxP sequences are localized in the N- and C-termini of CD20, respectively. The binding of the dimeric 14-3-3 adapter protein to these sequences can alter the intramolecular conformation of the two cytosolic tails of CD20 or stabilize CD20 homodimer or multimer formation. Alternatively, the 14-3-3 dimer could also connect CD20 to other signaling molecules. Using a 1-PLA and immunoprecipitation assays, we confirmed that GEF-H1, a GDP/GTP exchange factor that carries a conserved RxxSxP motif at its unstructured C-

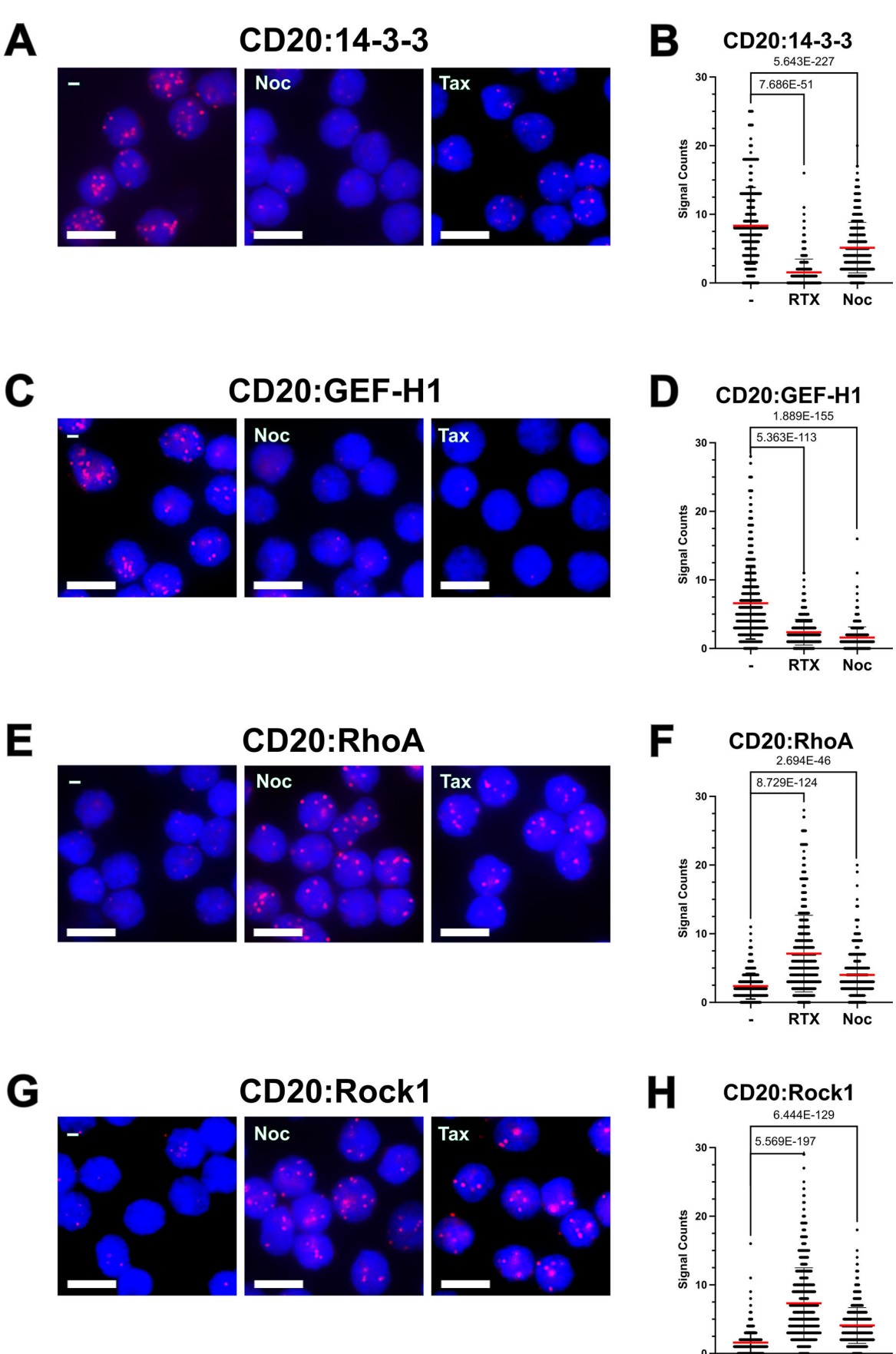

**Figure 6. Altered CD20 association in RTX or nocodazole-treated HD B cells.**

1-PLA studies of CD20 proximities in HD B cells, unstimulated (left), RTX-exposed for 5 min (middle), or treated for 1 h with Noc (right), $n = 3$. (**A**) 1-PLA study of CD20:14-3-3, (**B**) Quantification of (**A**); (**C**) 1-PLA study of CD20:GEF-H1, (**D**) Quantification of (**C**); (**E**) 1-PLA study of CD20:RhoA; (**F**) Quantification of (**E**); (**G**) 1-PLA study of CD20:ROCK1; (**H**) Quantification of 1-PLA data, all with CellProfiler. All PLA experiments were performed at least $n = 3$. PLA dots in red and nuclei stained in blue by DAPI, Scale bar 10 µm. PRISM10 was used to calculate significant differences in mean values quantification between the two groups using an unpaired Student's $t$-test, all exact p values were calculated with excel =T.DIST. Mean values with SD are presented. Source data are available online for this figure.

terminus, is one of the binding partners of the constitutively phosphorylated CD20. The serine residue S886 within the conserved RxxSxP motif of GEF-H1 is constitutively phosphorylated by Pak2 (Kosoff et al, 2013; Meiri et al, 2014). This modification is associated with the inhibition of the GDP/GTP exchange activity. Thus, two serine/threonine kinases, PKCδ and Pak2, control the formation and silencing of the CD20/GEF-H1 complex.

The activity of GEF-H1 is regulated by the cytoskeleton (Krendel et al, 2002). Specifically, GEF-H1 is intimately associated with the MT network, and this association inhibits its GDP/GTP exchange activity (Joo and Olson, 2021). There are two (non-mutually exclusive) mechanisms, by which MT binding inhibits GEF-H1. The first mechanism involves polymerized MT stabilizing the autoinhibited conformation, in which the N-terminal C1 domain folds over the PH domain of GEF-H1 (Jiang et al, 2016). The second mechanism is the binding of the light chain protein Tctex-1 of the dynein motor complex to the RRxxR binding motif in the unstructured C-terminal tail of GEF-H1 (Joo and Olson, 2021; Meiri et al, 2014). In this way, the dimeric dynein motor complex couples GEF-H1 to polymerized MT. The RxxSxP motif connecting GEF-H1 to the phosphorylated CD20 tails is located 60 amino acids downstream of the Tctex-1 binding site. Thus, in resting B cells, the autoinhibited GDP/GTP exchange factor functions as an adapter, connecting CD20 to the MT network.

## CD20 can orchestrate the MT-to-actin switch during B cell activation

In migrating, polarized B lymphocytes, the MT organization center (MTOC) is located near the trailing edge, where the MT network is involved in the organization of the uropod (Fais and Malorni, 2003). In polarized B cells, we found that the IgD-BCR nanocluster is localized at the uropod, whereas the IgM-BCR is associated with microvilli structures formed by core bundles of actin filaments at the leading edge (Saltukoglu et al, 2023). The interaction of CD20 with the MT network may facilitate the proper localization and function of the IgD-BCR nanocluster (Mattila et al, 2016). Interestingly, the IgD-BCR nanocluster harbors several receptors, such as BAFF-R and CD19, which carry a RRxxR or a related RRxxK motif for Tctex-1 binding in their cytosolic juxtamembrane region. Thus, multiple contact points seem to connect the IgD-BCR nanocluster to the MT network (J. Wang et al, 2018).

In polarized lymphocytes, RhoA and ROCK1 are also found at the uropod, and a RhoA deficiency or an inhibition of ROCK1 kinase activity prevents the formation of this structure (Sanchez-Madrid 2009). RhoA undergoes geranylgeranylation at its C-terminal CAAX motif, constitutively binding it with its lipid anchor to the inner leaflet of the plasma membrane (Roberts et al, 2008). Despite its permanent membrane association, our PLA studies of resting B cells did not detect RhoA in the vicinity of CD20 and the IgD-BCR nanocluster. Only upon MT destabilization and the following dissociation of the GEF-H1/CD20 complex, RhoA and ROCK1 is found in the vicinity of CD20. Interestingly, ROCK1 carries an RxxS motif near its C-terminus, suggesting that the GEF-H1/CD20 complex in resting B cells may be replaced by a RhoA-GTP/ROCK1/CD20 complex in activated B cells. GEF-H1 activation and liberation from the MT network could have a dual impact on ROCK1 activity and membrane location. The increased GDP/GTP exchange activity of GEF-H1 generates the binding partner of ROCK1, RhoA-GTP, while the GEF-H1-deprived CD20/14-3-3 complex is free to bind to ROCK1. These binding events counteract ROCK1 autoinhibition, localizing active ROCK1 at the IgD-BCR nanocluster. There, ROCK1 phosphorylates the myosin light chain (among other substrates), resulting in the formation of myosin-II filaments and actomyosin stress fibers with increased contractility (Lehtimäki et al, 2021). Stress fiber formation in polarized, antigen-exposed, activated B cells may be associated with uropod retraction, the dissociation of the IgD-BCR nanocluster and the observed movement of CD20 and CD19 to the IgM-BCR. According to this scenario, CD20 acts as both, a gatekeeper of the resting state and an actomyosin bundle-associated transporter that promotes the immunoreceptor coupling and organization motif (ICOM)-dependent conjugation of the IgM-BCR with the CD19 coreceptors in activated B lymphocytes.

## The effect of anti-CD20 antibodies on cytoskeleton reorganization

It was previously thought that anti-CD20 antibody binding did not directly affect the physiology of the targeted B cells, but rather simply flagged them for destruction via phagocytosis or ADCC mechanisms (Shanehbandi et al, 2017). However, several studies have indicated that anti-CD20 antibodies increase the susceptibility of targeted B cells to apoptosis. We found that exposing B cells to anti-CD20 antibodies results in the dissolution of MTs and MT instability, both of which are associated with apoptosis (Deans et al, 2002). Furthermore, caspase 3, an important effector of the apoptosis program, can cleave the inhibitory C-terminal domain, leading to a constitutively active form of ROCK1. Interestingly, this cleaved, active ROCK1 fragment retains the C-terminal RxxS motif and can still persist and associate with CD20. Thus, the N-terminal active kinase domain of ROCK1 remains bound to the plasma membrane in a RhoA-GTP-independent manner. A recent three-dimensional superresolution study has localized the RTX-engaged CD20 on actin bundles within microvilli structures (Ghosh et al, 2025; Kozlova et al, 2020). These findings are consistent with our observations that the IgM-BCR associates with an actin-based structural network of microvilli ridges and that CD20 is found in the vicinity of the IgM-BCR after exposure to either cognate

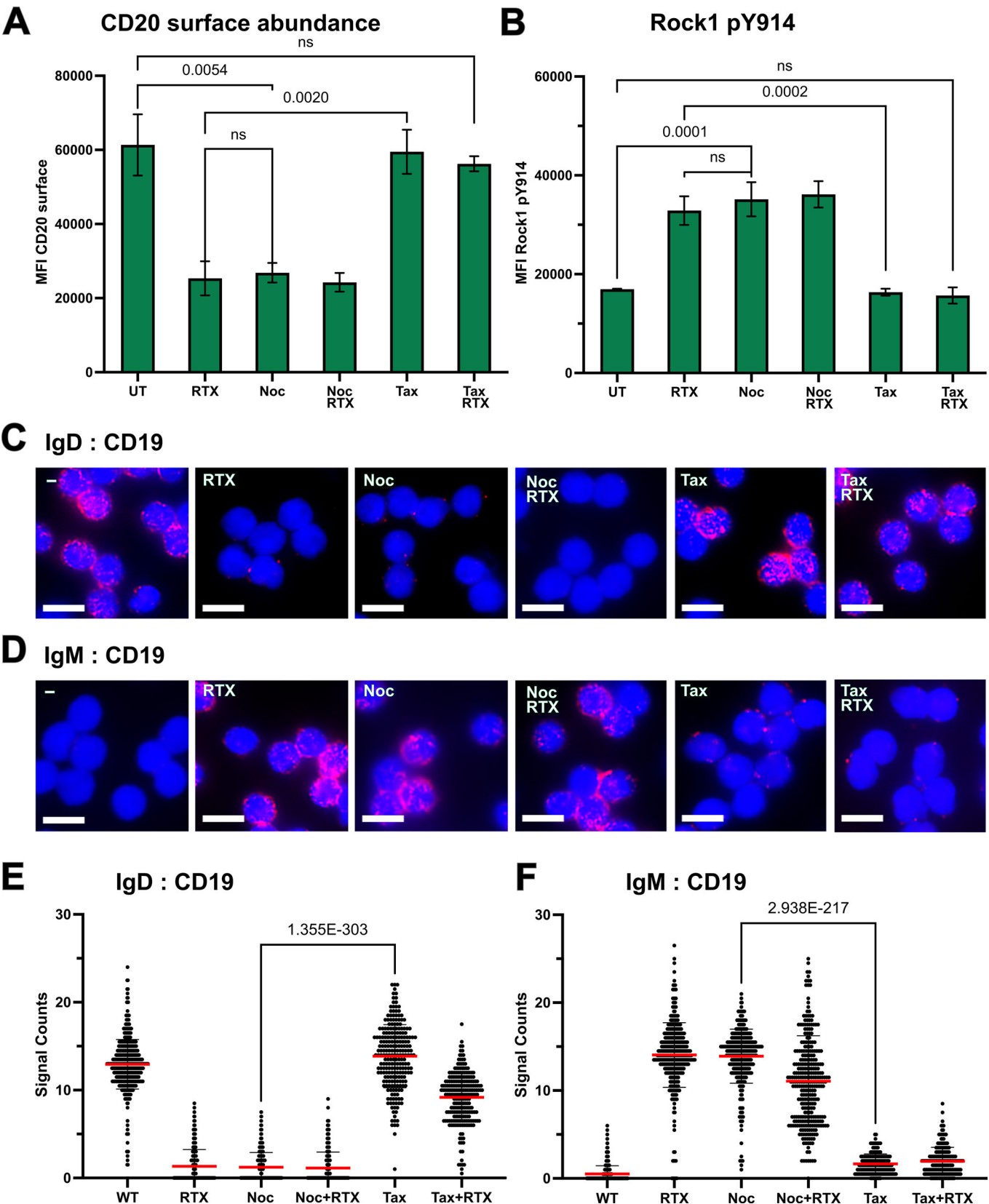

**Figure 7.  Stabilization of the IgD-BCR nanocluster and CD20 by the MT network.**

(**A**) CD20 expression on the surface of untreated (UT) HD B cells or after RTX-stimulation for 5 min. As indicated, some of these HD B cells were pretreated for 1 h with either nocodazole (Noc) or docetaxel (Tax). Shown is a summary of the MFI data of ≥3 flow cytometry experiments. PRISM10 students *t*-tests were used to calculate the significant differences between the experimental groups, n = 3. (**B**) Y914 phosphorylation and activation of Rock1 in untreated (UT) HD B cells or B cells stimulated for 5 min with RTX. As indicated, HD B cells were kept untreated, or pretreated for 1 h with nocodazole (Noc), or docetaxel (Tax). A summary of the MFI data of n ≥ 3 flow cytometry experiments is presented. Students *t*-tests were used to calculate the significant differences between the groups. Fab-PLA study of (**C**) the IgD-BCR:CD19 or (**D**) the IgM-BCR:CD19 proximity of HD B cells, kept untreated or stimulated for 5 min with RTX without or with pretreatment of the B cells with either Noc or Tax for 1 h as indicated. PLA dots in red and nuclei stained blue by DAPI, Scale bar 10 μm. (**E, F**) Quantification of Fab-PLA data with CellProfiler. All PLA experiments were performed at least n = 3. Significant differences between mean values of groups were calculated with PRISM 10 unpaired Student's *t*-test, the exact p values were calculated with excel =T.DIST. Mean values with SD are presented. Source data are available online for this figure.

antigen or anti-CD20 antibodies. Together, these data support assigning CD20 as a switch factor that interacts with the MT and actomyosin cytoskeleton in resting and activated B lymphocytes, respectively. The connection to actin bundles supports the rapid internalization of RTX-engaged CD20, and this may be an important factor for the design and success of anti-CD20 therapy (Lim et al, 2011; Vaughan et al, 2014, 2015; Williams et al, 2013). Further research is needed, but our study establishes the cellular function of CD20 as an adapter connecting antigen receptors to the underlying cytoskeleton, so CD20 is no longer an enigma of B-cell biology.

## Methods

### Reagents and tools table

| Cells | | |
|---|---|---|
| Healthy donor B cells | Uniklinik Freiburg | Buffy coats of transfusion medicine |
| Ramos Burkitt lymphoma | American Type Culture Collection | CRL-1923 |
| Phoenix cells | American Type Culture Collection | CRL-3213 |
| HEK293T cells | American Type Culture Collection | CLR-3216 |
| **Antibodies** | | |
| Anti-CD20 mc 2H7 | BioLegend | 302314 |
| Anti CD19 mc HIB19 | BioLegend | 302216 |
| Anti CD32b mc S18005H | BioLegend | 398306 |
| Anti IgM mc MHM88 | BioLegend | 314536 |
| Anti IgD mc IA6-2 | BoLegend | 348222 |
| Anti CD83 mc HB15e | BioLegend | 305326 |
| Anti CD22 mc HIP22 | BioLegend | 302514 |
| Anti CD45 mc HI30 | BioLegend | 304016 |
| Anti CD69 mc FN50 | BioLegend | 310912 |
| Anti GEF-H1 mc 55B6 | Cell Signaling | 4076 |
| Anti Rock1 pc | Invitrogen | PA5-81540 |
| Anti 14-3-3 pc | Proteintech | 14881-1-AP |
| Anti pGEF-H1 S886 pc | Invitrogen | PA5-105221 |
| Anti-pRhoA S188 pc | BIOSS | Bs 5330 R |
| Anti-RhoA mc 67B9 | Cell Signaling | 2117 |

| Cells | | |
|---|---|---|
| Anti-pRock1 Y914 pc | Invitrogen | PA5-105054 |
| Anti-CD27 mc M-T271 | BioLegend | 356412 |
| Anti-CXCR4 mc 12G5 | BioLegend | 306514 |
| Anti-HLA1 mc W6/32 | BioLegend | 311402 |
| Anti-β-actin mc 13E5 | Cell Signaling | 4970 |
| Anti-a-tubulin pc | Sigma-Aldrich | T5168 |
| Anti-mouse-A555 IgG mc | Invitrogen | A32727 |
| Anti-rabbit pc-HRP | Cell Signaling | 7076 |
| Anti-mouse pc-HRP | Cell Signaling | 7074 |
| Anti-PKCdelta pc | Cell Signaling | 2058 |
| Anti-flag mc D6W5B | Cell Signaling | 14793 |
| Anti-flag mc M2 | Sigma | F3165 |
| Phospho-PKCδ (Thr505) Antibody pc | Cell Signaling | 9374 |
| **Oligonucleotides and other sequence-based reagents** | | |
| Geneblock CD20 N-term | Intergrated DNA technologies (IDT) | 397 bp, see ET1 |
| Geneblock CD20 C-term | IDT | 258 bp, see ET1 |
| Geneblock CD20 wt | IDT | 482 bp, see ET1 |
| CD20 KO Exon 3 | IDT | gccctattgctatgcaatctgg |
| pLenti:PKCdelta | Addgene | Plasmid #52961 |
| CD32b KO Exon 4 | IDT | agcacgttgatccactgggg |
| pGEX-4T1-RhoA G17A | Addgene | Plasmid #69357 |
| **Chemicals, Enzymes and other reagents** | | |
| Proximity ligation assay, probemaker, plus, minus | Sigma-Aldrich | Duo92009-1KT Duo92010-1KT |
| PLA detection in situ orange | Sigma-Aldrich | Duo92007-100RXN |
| Fc blocker | BioLegend | 422302 |
| Glutathione Agarose | Thermo Fisher | 25237 |
| ECL Solution Femto SS | Thermo Fisher | 34096 |
| Rituximab | Hoffmann-La Roche-AG | CH |
| docetaxel trihydrate RP-56976 | MedChemExpress | 114977-28-5 |
| Nocodazole R17934 | Selleck-Chemicals | 2775 |
| LipoJet™ In Vitro Transfection Kit | SignaGen | Cat #: SL100468 |

| Cells | | |
|---|---|---|
| **Software and Database** | | |
| Prism 10 | GraphPad | Dotmatics |
| CellProfiler 4.2.5 | CellProfiler | https://cellprofiler.org/ |
| Software ZEN Black | ZEISS | 2.3 SP1 FP3 |
| Inkscape 1.4.2 | Inkscape | https://inkscape.org/ |
| Snapgene 4.3.11 | Snapgene.com | Dotmatics |
| ImageStudioLite | LICOR | https://licorbio.com/image-studio-lite |
| FlowJo | Tristar | https://www.flowjo.com/ |
| MaxQuant version 2.6.7.0 | MaxQuant | MaxQuant.org |
| Limma package, | R | https://www.r-project.org/ |
| clusterProfiler package version 4.12.6 | R | https://www.r-project.org/ |
| edge R, version 4.4.0 | R | https://www.r-project.org/ |
| PRIDE data | ProteomXchange | https://www.proteomexchange.org/ |
| Molecular Signatures Database MSigDB, version 7 | GSEA | Broad Institute |
| GEO accession GSE300605 | NCBI | https://www.ncbi.nlm.nih.gov/geo/query/acc.cgi?acc=GSE300605) |
| **Other** | | |
| Airyscan Zeiss | ZEISS | |
| DMi8 microscope | LEICA | |
| ChromoTek DYKDDDDK Fab-Trap® Agarose | Proteintech | Cat No. ffa |
| Attune NxT Flow Cytometer | Thermo Fisher Scientific | |
| UltiMate 3000 RSLCnano HPLC system | Thermo Fisher Scientific | Dreieich, Germany |
| QExactive Plus mass spectrometer nanoEase M/Z Symmetry C18 Trap | Thermo Fisher Scientific | Bremen, Germany, Waters Corporation, Milford, MA |
| Q-page tgn percast gel (12 wells; 10%), smobio | 7Biosciences | QP5220 |
| IBIDI uncoated 8-well high glass-bottom chambers | IBIDI | Cat. No 80807 |
| Nanospray Flex ion source with DirectJunction adapter | Thermo Fisher Scientific | |
| PTFE slides | Fisher Scientific | NC9811708 |
| Qubit 3.0; | Life Technologies | |
| RNeasy Plus Kit | Qiagen | cat# 74143 |
| NEBNext® Ultra™ II Directional RNA Library Prep Kit | NEB | cat # E7760L |
| NextSeq® 500/550 High Output Kit v2 | Illumina | Cat. FC-420-1001-4 |
| Mycostrip® | Mycoplasma Detection Kit | Highly specific | Invivogen | Cat. rep-mys-10 |

## Methods and protocols

### Human naïve B cells

Primary naïve B cells were obtained from fresh buffy coats, provided by the Institute for Transfusion Medicine and Gene Therapy, ITG, Freiburg. Peripheral blood mononuclear cells (PBMCs) were separated by Ficoll gradient centrifugation of whole blood and negatively selected using the EasySep Human Naïve B Cell Isolation Kit (STEMCELL). Prior to experiments, living cells were determined and counted (with trypan blue) and negatively selected using the EasySep Human Naïve B Cell Isolation Kit (Stemcell). Prior to experiments, primary B cells were controlled for purity by staining of CD19, IgM, IgD, CD32b, and CD20 and rested overnight, at least for 6 h. The study was approved by the Institutional Review Board of the University of Freiburg (Ethical votes No 507/16 and 336/16).

### Cell culture

The human Burkitt lymphoma B cell line Ramos was obtained from American Type Culture Collection (ATCC, Ramos cat. CRL-1923, RRID: CVCL 1646) and stably transfected with ecotropic receptor (EcoR) to allow murine retroviral transfection with MMLV particles. Ramos B cells were cultured in RPMI 1640-Glutamaxx medium, 10% FCS (PAN, Biotech), 10 units/mL penicillin/streptomycin (Gibco), 20 mM Hepes (Gibco), and 50 μM β-mercaptoethanol (Sigma) in a humidified saturated atmosphere at 37 °C with 5% $CO_2$.

Phoenix cells were cultured in six-well plates (Greiner-one) at 37 °C with 7% $CO_2$ in 2–3 ml RPMI 1640 + GlutaMAX supplemented with 10% heat-inactivated FCS, 100 U/ml penicillin/streptomycin, and 50 μM 2-mercaptoethanol at about 50% confluency. For splitting, the cell medium was removed, and 0.5 ml trypsin was added. After a short incubation time (about 5 min), cells were washed with a high volume of PBS. Finally, some cells (1/10 dilution) were transferred into fresh culture medium.

HEK 293T cells, human embryonic kidney cells, were cultured in complete DMEM GlutaMAX medium supplemented with 10% FCS, 10 mM HEPES, 10 μM sodium pyruvate, 50 units/mL penicillin, and 50 μg/mL streptomycin in a humidified saturated atmosphere at 37 °C with 7.5% $CO_2$. All cells were routinely tested for mycoplasma contamination.

### Labeling for quantitative phosphoproteomic study

For the quantitative phosphoproteomic study, Ramos B cells were labeled using stable isotope labeling by amino acids in cell culture (SILAC). Light labeling was performed using 12C6-L-arginine/12C6-L-lysine. For medium-heavy and heavy labeling, 13C6-L-arginine/D4-L-lysine and 13C615N4-L-arginine/13C615N2-L-lysine were used, respectively.

### Generation of the Ramos CD32b (FCGR2B) KO and CD20 (MS4A1) KO cell lines

The human genes for CD32b (*FCGR2B*) and CD20 (*MS4A1*) were rendered defective in the Ramos B cell system using the CRISPR/Cas9 system. Synthetic single guide RNAs (sgRNA)s targeting defined 20-bp sequences within the corresponding coding regions (agcacgttgatccactggggg for CD32b KO located in exon 4 of *FCGR2B* and gccctattgctatgcaatctgg within exon 3 of MS4A1 for CD20 KO,

respectively) were purchased from Integrated DNA Technology (IDT). The CRISPR-Cas components were delivered to the Ramos B cell line in the form of ribonucleoprotein (RNP) complex, using the above mentioned sgRNAs pre-assembled with Cas9 protein (IDT) via the NEON Transfection System (Invitrogen) following the manufacturer's instructions. Electroporation for Ramos cells was performed in 10 μL NEON tips at 1350 V, 30 ms, single pulse. As a control, one sample without crRNA was prepared and electroporated in parallel. The transfected cells were first recovered for 72 h at 37 °C and 5% $CO_2$ without antibiotics and then subjected to complete RPMI medium. Successfully transfected Ramos cells were batch sorted using a Bio-Rad cell sorter in three different rounds and monitored throughout experiments. Inactivation of the target genes was verified by flow cytometry.

### Generation of the Ramos PKCδ (PRKCD) gene KO

The sequences AACCCAATCATAGCAGAGC or GCTGAGTT-CAGTGAGTGC, each targeting the *PRKCD* coding region, were inserted into a lentivirus CRISPR vector (Addgene) to generate the corresponding sgRNA cassettes. The constructs were co-transfected with an mCherry expression vector into Ramos cells. The cells were incubated at 37 °C for 24 h in a humidified $CO_2$ incubator. The following day, single cells were sorted for mCherry reporter gene expression with the BD FACSAria™ III Cell Sorter. The sorted single cells were cultured for 3–4 weeks in complete RPMI medium supplied with 10% FBS in a humidified $CO_2$ incubator at 37 °C. The expected *PRKCD* gene deficiency was verified by Sanger sequencing and the loss of PKCδ protein expression by Western blotting. The generated cell line was independently re-analysed.

### Lentiviral transfection for the generation of CD32b-tr Ramos B cells

The human embryonic kidney HEK 293T cells were cultured in complete DMEM GlutaMAX medium supplemented with 10% FCS, 10 mM HEPES, 10 μM sodium pyruvate, 50 units/mL penicillin, and 50 μg/mL streptomycin in a humidified saturated atmosphere at 37 °C with 7.5% $CO_2$. Lentivirus particles were obtained by co-transfecting HEK 293T cells with pLenti-hCD32b-IRES-GFP, pCMVΔR8.74, and pMD2vsvG plasmids using polyethyleneimine (PEI, Polysciences). After 24 and 48 h of incubation and post-transfection, viral supernatant was harvested, sterile filtered and combined. Lentiviral particles were enriched by centrifugation (4 h, 10,000 × g, 8 °C) after placing them in a 1:5 ratio on a 10% sucrose layer. The lentiviral pellet was resuspended in DMEM Glutamaxx without supplements and stored at –80 °C. The viral titers were assessed by determining the multiplicity of infection (MOI). The GFP expression was analyzed by flow cytometry, and GFP-positive cells were batch sorted three times. The lentiviral titer was calculated: Transduction units/mL = (number of cells × % of GFP$^+$ cells × dilution factor)/(mL of lentivirus dilution). CD32b expression was monitored throughout the experiments.

### Construction of WT or S/A mutated CD20-flag expression vectors

The CD20-flag vector was modified by GeneBlocks (IDT) encoding the N-terminal or C-terminal tails with S/A exchanges at the chosen four positions, respectively. Geneblocks were solved in IDTE buffer according to the manufacturer's protocol and subsequently inserted into the pMIG-CD20-flag-IRES-GFP vector using infusion cloning method. Plasmids were sequenced (GATC, Eurofins) and used to generate the CD20WT, CD20Nmut, CD20Cmut, and CD20NCmut Ramos cell lines (CD20 geneblocks in Table EV1).

### Retroviral transfection of CD20 mutants

Murine retrovirus-containing supernatants were obtained by transfecting Phoenix-eco cells with LipoJet (SignaGen) according to the manufacturer's protocol and different pMIG-CD20-IRES-GFP constructs, or an empty GFP-control plasmid. Therefore, $4 \times 10^5$ phoenix cells were plated into six-well plates and incubated for 24 h. Old medium was soaked off and 1 mL fresh medium was added. For transfection, 3 μL Gene Juice (or Jet PEI solution) was added to 100 μL RPMI without FCS, vortexed and incubated at RT for 5 min. About 1 μg plasmid DNA was added and incubated at RT for 5–15 min until droplets have been formed. An additional 0.5 μg pkat plasmid was added, and the transfection solution was dropped onto the cells. After 48 h, viral supernatant was collected, sterile filtered and used directly or stored in −80 °C. For transduction, $4 \times 10^5$ newly generated CD20 KO Ramos cells were resuspended in 1 mL of viral supernatant containing Polybrene (1 μg/mL) and spin-infected by centrifugation (180 min, 300 × g, 37 °C). The viral supernatant was removed, and cells were cultured and batch sorted for GFP expression in three rounds (Bio-Rad cell sorter).

### RNA sequencing

Ramos wt or PKCδKO Ramos were cultured with or without IgM (10 ug/ml) stimulation for 4 h, and then RNAs were extracted by RNeasy plus kit (Qiagen cat# 74143). RNA libraries for RNAseq were prepared using the NEBNext® Ultra™ II Directional RNA Library Prep Kit for Illumina (NEB cat # E7760L) according to the manufacturer's instructions, and samples were sequenced 84-bases long on the NextSeq® 550 with high output flowcell (NextSeq® 500/550 High Output Kit v2).

### GEO data

Downstream analyses were conducted using R (version 4.4.0). Raw read counts were normalized for library size and composition using the trimmed mean of M-values (TMM) method. Differential gene expression analysis comparing the resting PKCδ ko and wt cell conditions was performed with the edgeR package (version 4.2.2) (Chen et al, 2025). Gene set enrichment analysis was carried out using the clusterProfiler package (version 4.12.6) (Wu et al, 2021) with gene sets from the Molecular Signatures Database (MSigDB, version 7) (Subramanian et al, 2005). For all statistical analyses, significance was defined as an adjusted *p* value <0.05.

### Flow cytometry analysis and phospho-flow detection

For surface staining, $1–20 \times 10^5$ cells were Fc-blocked (Human TruStain, BioLegend), stained with fluorescent antibodies on ice, washed twice, and then their fluorescence was measured with a FACS Attune NxT (Life Technologies). For intracellular staining and phospho-flow analysis, the cells were activated for the requested time points, immediately fixed with 4% PFA, washed twice and permeabilized with 0.5% saponin (Quillaja bark, Sigma) in FACS buffer containing protease and phosphatase inhibitors (HALT, Thermo Fisher) for 30 min. Samples were washed and stained overnight at 4 °C. If necessary, secondary antibody staining was performed, and the background was controlled using only the secondary antibody. Data were exported in FCS-3.0 format and analyzed with FlowJo software (TreeStar).

### Western blotting

For western blotting unstimulated or activated Ramos B cells were counted, collected, washed once with PBS, and immediately lysed in ice-cold 100 µL RIPA buffer (10 mM Tris/Cl, pH 7.5, 150 mM NaCl, 0.5 mM EDTA, 0.1% SDS, 1% Triton™ X-100, 1% deoxycholate (pH adjusted at +4 °C) supplemented with protease and phosphatase inhibitors (Cocktail 100X (Thermo #78440) for 30 min on ice. Optional, the lysates were sonicated for 15 s, $3 \times 10\%$ with 50% power and centrifuged for 15 min at $14,000 \times g$ in an ice-cold centrifuge. Equal amounts of adjusted and cleared lysates (Qubit 3.0; Life Technologies) were subjected to SDS–PAGE on 10% mini precast gels (7Bioscience). After transfer, the PVDF membrane (GE Healthcare Amersham) was blocked with 5% BSA in PBS and 0.1% Tween 20 with intensive washing steps with PBS-T in between. The membrane was first exposed to primary antibodies specific for the proteins under study (CD20, 14-3-3 or GEF-H1). A horseradish (HRP) peroxidase-conjugated goat anti-rabbit or goat anti-mouse antibody was used as a secondary antibody before detection with ECL chemiluminescent substrate (Bio-Rad).

### Affinity purification of CD20-flag and GEF-H1

The anti-flag immunoprecipitation was performed according to the manufacturer's protocol (Chromotek). In brief, $20 \times 10^6$–$10^7$ Ramos cells were counted, collected, and washed once with PBS. The cells were subjected to 37 °C and kept unstimulated or activated (RTX) for the requested time points (5–8 min). Then cells were placed on ice, washed once with ice-cold PBS and centrifuged for 5 min at 4 °C at $300 \times g$ in a precooled centrifuge. Cell pellets were lysed by extensive pipetting on ice for 30 min in 200 µL in ice-cold lysis buffer (10 mM Tris/Cl, pH 7.5, 150 mM NaCl, 0.5 mM EDTA, 0.5% IGEPAL (pH was adjusted at +4 °C) supplemented with a protease/phosphatase inhibitor cocktail (HALT, Thermo Fisher). Lysates were centrifuged for 10 min at $17,000 \times g$ and diluted in 200 µL dilution buffer (10 mM Tris/Cl, pH 7.5, 150 mM NaCl, 0.5 mM EDTA, pH was adjusted at +4 °C) supplemented with a protease/phosphatase inhibitor cocktail. About 50 µL of total lysate were kept for the input analysis. Diluted lysates were rotated overnight with equilibrated anti-flag agarose beads. Beads were then sedimented by centrifugation ($2500 \times g$, 4 °C, 5 min) and washed extensively. Bound proteins were eluted with 2x SDS–PAGE Laemmli buffer, and samples were boiled at 95 °C for 5 min and subjected to SDS–PAGE, followed by Western blotting.

RhoA(G17A), the nucleotide-binding deficient mutant of RhoA, binds strongly to GEF-H1 and can be used for GEF-H1 purification and quantification (Waheed et al, 2012). For this, a glutathione S-transferase (GST) coupled RhoA(G17A) fusion protein was produced in *E. coli* bacteria, and the GST-RhoA(G17A) was loaded on glutathione Sepharose beads. The GST-RhoA(G17A)-beads were mixed with Ramos B cell lysate and rotated overnight at 4 °C. Beads were washed extensively with lysis buffer, boiled for 5 min and the cleared supernatant subjected to SDS–PAGE and western blotting.

### Proximity ligation assay (PLA)

The Fab-PLA was performed as described earlier (Kläsener et al, 2018). In brief: The antibodies used for the PLA were selected based on clonality and concentration. They should also be free of BSA and other supplements such as carriers, or preservatives. For Fab-PLA, the F(ab)- fragments were prepared from the corresponding antibodies using the Pierce Fab Micro Preparation Kit (Thermo

Fisher). The produced F(ab)-fragments or primary antibodies (1-PLA) were coupled with PLA probemaker (Sigma-Aldrich) after buffer exchange (Zeba™ spin desalting columns, Thermo Fisher). All PLA experiments were performed as technical triplicates. Ramos or HD naïve B cells were settled on polytetrafluoroethylene (PTFE) slides (Thermo Fisher Scientific) for 30 min at 37 °C and kept unstimulated or activated with RTX [10 µg/mL] for 5 min and then fixed for 20 min with 4% paraformaldehyde.

After extensive washings, the slides were blocked for 30 min with blocking buffer (25 µg/mL sonicated salmon sperm DNA, 250 µg/mL BSA, and 1 M glycine). PLA was performed with the Duolink In Situ Orange (Sigma-Aldrich). Slides were directly mounted with DAPI Fluoromount-G (Southern Biotech) to visualize the PLA signals in relation to the nuclei. For each experiment, a minimum of 1000 HD B cells or Ramos cells were analyzed with CellProfiler 3.0.0. In brief, the metadata of images were exported, and the intensity of the input frames was rescaled to the full intensity range in the RGB scheme. PLA signals (size of PLA dots 4–10 pixels) at a distance of 10 pixels to the Nuclei (size 15–80 pixels) with a threshold of 0.0075 in at least three different sample areas were measured and subjected to analysis. Images of the PLA showing dead cells, cell fragments, or non-specific accumulation of fluorophores were excluded from the analysis. One representative assay of at least 1000 HD B cells or 100 Ramos cells is presented from all collected and analyzed data.

### Imaging, image analysis, and data processing

All microscope images for PLA were acquired using a Leica DMi8 microscope equipped with an HC PL APO 63×/1.40–0.60 oil immersion objective lens and analyzed with CellProfiler 3.0.0. and Prism software (GraphPad, La Jolla, CA).

### Microtubule staining

MT staining was performed as described in (J. C. Wang et al, 2018). For MT staining, Ramos B cells were settled on uncoated glass-bottom chambers (IBIDI) for 30 min at 37 °C, kept unstimulated or activated with RTX for 20 min, washed once with PBS and then fixed for 20 min with 4% paraformaldehyde and 0.1% glutaraldehyde. After washing, blocking and permeabilization with 250 µg/mL BSA, 1 M glycine, 0.5% saponin for 30 min, cells were stained with anti-α tubulin (Sigma-Aldrich) overnight, washed and stained with secondary anti-mouse Alexa fluor-555 antibody (Invitrogen). Fixed Ramos cells in IBIDI chambers were kept in PBS until microscope imaging.

HD B cells were settled on PTFE slides (Thermo Fisher) for 30 min at 37 °C, kept unstimulated, or activated with RTX [10 µg/mL] for 5 min, with or without pretreatment with Tax for 1 h and then fixed for 20 min with 4% paraformaldehyde and 0.1% glutaraldehyde. After washing, blocking and permeabilization with 250 µg/mL BSA, 1 M glycine, 0.5% saponine for 30 min, cells were stained with anti-α tubulin (Sigma-Aldrich) overnight, washed and stained with secondary anti-mouse Alexa fluor-555 antibody (Invitrogen). Slides were directly mounted with Fluoromount-G mounting-solution (Invitrogen) and subjected to microscope analysis.

### Microscopy

The high-resolution visualization of MTs was performed using an LSM 880 Airyscan laser scanning confocal microscope attached to an inverted microscope Axio Observer Z1 (Carl Zeiss Microscopy).

The Alexa Fluor-555 labeled α-tubulin was excited with 561 nm, and the emission was detected with the Airyscan detector in SuperResolution (SR) mode. For all cells, z-stacks were acquired with a Plan-Apochromat 63x/1.40 Oil DIC M27 (Appendix Fig. S6) or a LD LCI Plan-Apochromat 40x/1.20 autocorr (Appendix Fig. S5) objective. The immersion medium was oil (63x) or water (40x).

Zeiss Software ZEN Black 2.3 SP1 FP3 (release 14.0.22.201) was used for acquisition. (acquisition settings for Appendix Fig. S6: EM filter SP615, pixel size xy:0.049 μm, z-step size: 0.199 μm, pixel dwell time 1 μs, bi-directional scanning, for Appendix Fig. S5: no EM filter, pixel size xy:0.049 μm, z-step size: 0.243 or 1 μm, pixel dwell time 1.92 μs, bi-directional scanning). All images were processed with the above mentioned ZEN Black software using the Airyscan algorithm and with Zen Blue (Version 3.9.3) for subset and maximum intensity projection. Airyscan processed images shown in Appendix Fig. S5 are single z-planes from an image z-stack, and in Appendix Fig. S6, maximum intensity projections of the full image z-stack. The display gain is set to 0.75 and the intensity value to Best fit.

### Statistical analysis

Students *t*-test was used for the experiments, if not otherwise described, to determine statistical significance provided by Prism 10 software (GraphPad, La Jolla, CA).

### MT network stabilization or destabilization

For the stabilization of MT HD naïve B cells or Ramos B cells were treated for 1 h with 100 nM docetaxel trihydrate (MedChemExpress) dissolved in DMSO and diluted in RPMI culture medium. The destabilization of MT cells were treated for 1 h with 10 μM nocodazole (Selleck-Chemicals), dissolved in DMSO and diluted in RPMI culture medium.

### Rituximab (RTX) treatment

RTX was kindly provided by F. Hoffmann-La Roche-AG. For naïve HD B cells or Ramos cell treatment, RTX was diluted to a final concentration of 10 μg/mL and experiments were performed at least three times.

### Phosphopeptide enrichment

SILAC-labeled Ramos B-cells (wt and PKCδ KO) were starved by serum depletion (FCS) for 30 min. Following starvation, cells were harvested by centrifugation at 1200 rpm for 5 min, washed twice with starvation medium. As a control, mock treatment was performed using PBS. After activation, cells were rapidly snap-frozen in liquid nitrogen to preserve their signaling status. Cell lysis was performed using GdmHCl buffer (6 M GdmHCl, 100 mM Tris-HCl, pH 8.5, 10 mM TCEP, and 40 mM chloroacetamide). The cell pellets were resuspended in the lysis buffer, followed by sonication (two cycles of 30 s). Protein denaturation was achieved by incubating the lysates at 95 °C for 5 min. To remove cellular debris, the lysates were centrifuged at $3500 \times g$ for 30 min at 4 °C. Protein concentration was determined using a Bradford assay. To precipitate proteins, four volumes of ice-cold acetone were added to the lysate. The protein precipitates were collected by centrifugation and washed with ice-cold acetone to remove residual contaminants. Proteins were digested using a combined Lys-C/trypsin digestion. Phosphopeptides were enriched using the Easyphos workflow. The phosphopeptide-enriched eluates were desalted using C18 StageTips (3M Company), pooled, and stored at −80 °C until further analysis.

### LC-MS/MS analysis

Peptide mixtures, reconstituted in 0.1% TFA, were analyzed by nano HPLC-ESI-MS/MS using an UltiMate 3000 RSLCnano HPLC system (Thermo Fisher Scientific, Dreieich, Germany) online coupled to a QExactive Plus mass spectrometer (Thermo Fisher Scientific, Bremen, Germany). The RSLC system was equipped with C18 trap columns (nanoEase M/Z Symmetry C18 Trap; 20 mm length, 180 μm inner diameter, 5 μm particle size, 100 Å pore size, Waters Corporation, Milford, MA) and an analytical C18 reversed-phase nano LC column (nanoEase M/Z HSS C18 T3; 250 mm length, 75 mm inner diameter, 1.8 μm particle size, 100 Å pore size, Waters Corporation, Milford, MA). A binary solvent system consisting of 0.1% (v/v) formic acid (FA) as solvent A and 80% (v/v) acetonitrile (ACN)/0.1% (v/v) FA as solvent B was employed for peptide separation. Peptide mixtures were loaded, washed and preconcentrated on the pre-column for 5 min using solvent A and a flow rate of 10 μL/min. A gradient was then applied at a flow rate of 300 nL/min ranging from 4 to 39% B in 140 min, 39–54% B in 15 min, 54–95% in 5 and 3 min at 95% B. Eluted peptides were transferred to a fused silica emitter for electrospray ionization enabled by a Nanospray Flex ion source with DirectJunction adapter (Thermo Fisher Scientific), applying a spray voltage of 1.6 kV and a capillary temperature of 250 °C. MS/MS data were acquired in data-dependent mode using the following parameters: MS precursor scans at m/z 370–1700 with a resolution of 70,000 (at m/z 400); automatic gain control (AGC) of $3 \times 10^6$ ions; a maximum injection time (IT) of 60 ms; a top 12 method for higher-energy collisional dissociation of multiply charged precursor ions with a normalized collision energy of 28%. MS/MS scans from 200 to 2000 m/z were recorded at a resolution of 35,000. The AGC for MS/MS scans was set to $1 \times 10^5$ with a maximum IT of 120 ms and a dynamic exclusion time of 45 s.

### MS data analysis

MaxQuant (version 2.6.7.0) with its integrated Andromeda search engine (Cox et al, 2011; Cox and Mann, 2008) was employed. MS/MS data were searched against the human proteome set with isoforms downloaded from UniProt (105300 entries). Protein identification was performed using MaxQuant default settings (including carbamidomethylation of cysteine residues as fixed modification and N-terminal acetylation and methionine oxidation as variable modifications), with the following exceptions: "Arg6;Lys4" and "Arg10;Lys8" were specified as medium and heavy modification labels, respectively; a maximum of three missed cleavages was allowed; "phospho (STY)", "oxidation (M)", and "acetyl (Protein N-term)" were selected as variable modifications; and the options 'match between runs' and "requantify" were activated.

Ratios between labels (normalized by MaxQuant) were extracted from the Phospho(STY)Sites.txt file, reverse entries and potential contaminants were removed, and the list was filtered for at least two unique peptides per phosphosite. Phosphosite ratios were normalized to the respective ratio at the protein group level using the median ratio of all protein group ratios referenced in the "Protein group IDs" column of the Phospho(STY)Sites.txt file. Resulting phospho site ratios were filtered so that only sites with valid values in ≥3 of 4 replicates were retained. Significance of

differential regulation was tested using linear models implemented in the LIMMA package, and the resulting fold changes and significance level were plotted. Data analysis was performed in Python 3.10 using the autoprot wrapper (Bender et al, 2024) for data analysis and plotting. All original code for MS data analysis has been deposited at Zenodo.

## Data availability

The mass spectrometry data from this publication have been deposited in the PRIDE database (www.ebi.ac.uk/pride/archive/projects/PXD063667) with the identifier PXD063667. The code used in this study for mass spectrometry data analysis and the results of this analysis have been deposited to Zenodo (https://doi.org/10.5281/zenodo.15350913). The RNAseq data from this publication have been deposited in the Gene Expression Omnibus database (https://www.ncbi.nlm.nih.gov/geo/query/acc.cgi?acc=GSE300605) and assigned the identifier GSE300605.

The source data of this paper are collected in the following database record: biostudies:S-SCDT-10_1038-S44318-026-00781-5.

## Peer review information

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

## Acknowledgements

We thank Dr. Christian Klein and the Roche Innovation Center Zurich for financial support and anti-CD20 antibodies, and Falk Nimmerjahn, University of Erlangen, for the CD32B plasmid. We would like to thank Dr. Baerbel Keller for her help with the RNA sequencing data. We thank Dr. Lise Leclercq for correcting the manuscript. This work was supported by the Roche research contract ZVK-2023101202 and an RO1 grant of the National Institutes of Health (NIH) under the award number A031503 (SMA, MR). The Lighthouse Core Facility is funded in part by the Medical Faculty, the University of Freiburg (Project Numbers 2023/A2-Fol; 2021/B3-Fol), the DKTK, and the DFG (Project Number 450392965). The Zeiss LSM 880 microscope was funded by the Deutsche Forschungsgemeinschaft (DFG) project No. 31778431, DFG grant WA 1597/3-1 to KW. Work in the Warscheid lab was supported by the DFG SPP2453 (project No. 541758684), FOR2743 (project 9) and TRR130 (project C01) for MR and BW.

## Author contributions

**Kathrin Kläsener**: Conceptualization; Resources; Data curation; Software; Formal analysis; Supervision; Funding acquisition; Validation; Investigation; Visualization; Methodology; Writing—original draft; Project administration; Writing—review and editing. **Cindy Eunhee Lee**: Conceptualization; Resources; Data curation; Formal analysis; Validation; Investigation; Methodology; Project administration. **Julian Bender**: Conceptualization; Resources; Data curation; Software; Formal analysis; Validation; Investigation; Visualization; Methodology; Writing—original draft; Project administration; Writing—review and editing. **Angela Naumann**: Conceptualization; Resources; Data curation; Software; Formal analysis; Validation; Investigation; Visualization; Methodology; Writing—original draft; Project administration. **Lena Reimann**: Conceptualization; Resources; Data curation; Software; Formal analysis; Validation; Investigation; Methodology; Writing—original draft. **Geoffroy Andrieux**: Conceptualization; Resources; Data curation; Software; Formal analysis; Validation; Investigation; Visualization; Methodology; Writing—original draft; Writing—review and editing. **Claudio Mussolino**: Conceptualization; Resources; Data curation; Software; Formal analysis; Validation; Investigation; Visualization; Methodology; Writing—original draft; Project administration. **Nadja Herrmann**: Conceptualization; Data curation; Investigation; Methodology. **Roland Nitschke**: Resources; Data curation; Software; Formal analysis; Supervision; Funding acquisition; Validation; Investigation; Visualization; Methodology; Writing—original draft. **Reinhard E Voll**: Resources; Funding acquisition; Validation; Investigation; Writing—review and editing. **Bettina Warscheid**: Resources; Funding acquisition; Validation; Investigation; Writing—review and editing. **Klaus Warnatz**: Resources; Funding acquisition; Validation; Investigation; Writing—review and editing. **Michael Reth**: Conceptualization; Supervision; Funding acquisition; Validation; Investigation; Visualization; Methodology; Writing—original draft; Project administration; Writing—review and editing.

Source data underlying figure panels in this paper may have individual authorship assigned. Where available, figure panel/source data authorship is listed in the following database record: biostudies:S-SCDT-10_1038-S44318-026-00781-5.

## Funding

## Disclosure and competing interests statement

The authors declare no competing interests.

