## [Peer Review File · The EMBO Journal]

CD20 tails interact with the 14-3-3/GEF-H1 complex and microtubule network upon PKC δ phosphorylation

Kathrin Kläsener, Cindy Lee, Julian Bender, Angela Naumann, Lena Reimann, Geoffroy Andrieux, Claudio Mussolino, Nadja Herrmann, Roland Nitschke, Reinhard Voll, Bettina Warscheid, Klaus Warnatz, and Michael Reth

Corresponding author(s): Michael Reth (michael.reth@bioess.uni-freiburg.de) , Kathrin Kläsener (kathrin.klaesener@bioess.uni-freiburg.de)

Review Timeline:

Submission Date:	30th Jul 25
Editorial Decision:	14th Oct 25
Revision Received:	8th Jan 26
Editorial Decision:	25th Feb 26
Revision Received:	13th Mar 26
Accepted:	19th Mar 26

Editor: Ioannis Papaioannou

Transaction Report:

Dear Dr. Kläsener,

Thank you again for the submission of your manuscript EMBOJ-2025-122022 for consideration by The EMBO Journal, and for your patience during peer review. Once again, I would like to sincerely apologize for the rather slow process -for the standards of our journal- on this occasion, but as I have already informed you we have now received a complete set of comments from the three referees who assessed your study (included below).

I am pleased to say that the referees indicate interest in your study, find your results novel, interesting, and relevant, and point out that the conclusions are largely supported by the experimental data. However, they also identify a number of limitations and provide constructive suggestions for strengthening the manuscript further and increasing its impact on the field.

In light of the positive referees' comments and recommendations, I would like to invite you to submit a revised version of your manuscript taking the referees' suggestions on board, along with a detailed point-by-point response addressing all referees' comments. Please note that it is The EMBO Journal policy to allow only a single round of major revision, and acceptance of your manuscript will therefore depend on the completeness of your responses in this revised version. I think it would be useful to discuss your revision plan already during the initial phase of your revision. You are very welcome to share with me a draft point-by-point response letter/revision plan explaining if there are any points you do not agree with or cannot address, or alternatively we could arrange a video call, if you prefer.

We generally allow three months as standard revision time (January 13, 2025). As a matter of policy, competing manuscripts published during this period will not negatively impact our assessment of the conceptual advance presented by your study. However, we request that you contact us as soon as possible upon publication of any related work, to discuss how to proceed. Should you foresee a problem in meeting this three-month deadline, please let us know in advance and we will be able to grant an extension.

Please let me know if you have any questions or comments that you would like to discuss with me. Thank you for the opportunity to consider your work for publication in The EMBO Journal. I look forward to your revision.

Best regards,

Ioannis

Instructions for preparing your revised manuscript

1. When you are ready to submit the revision, please upload:

- A Word file of the manuscript text (including legends of main Figures, EV Figures and Tables). Please make sure that changes are highlighted (or "tracked") to be clearly visible.

- Individual production-quality figure files (one file per figure). When assembling your figures, please refer to our figure preparation guidelines in order to ensure proper formatting and readability in print as well as on screen:

If the data shown in a figure are obtained from n {less than or equal to} 2, please use scatter plots showing the individual data points.

- i. the name of the statistical test used to generate error bars and P values
- ii. the number (n) of independent experiments (please specify technical or biological replicates) underlying each data point (discussion of statistical methodology can be reported in the Materials and Methods section, but figure legends should contain a basic description of n , P , and the test applied)
- iii. the nature of the bars and error bars (s.d., s.e.m.).

- A point-by-point response to the referees' comments, with a detailed description of the changes made (as a word file). All referees' concerns must be fully addressed and their suggestions taken on board. When preparing your letter of response to the referees' comments, please bear in mind that this will form part of the Review Process File and will therefore be available online to the community. Please note that you have the possibility to opt out of the transparent process at any stage prior to publication by letting the editorial office know (contact@embojournal.org); if you do opt out, the Review Process File link will point to the following statement: "No Review Process File is available with this article, as the authors have chosen not to make the review process public in this case.". For more details on our Transparent Editorial Process, please visit our website: <https://www.embopress.org/page/journal/14602075/authorguide#transparentprocess>

- Expanded View (EV) files (replacing Supplementary Information) that are collapsible/expandable online. A maximum of 5 EV Figures can be typeset. EV Figures should be cited as "Figure EV1, Figure EV2" etc. in the text, and their respective legends should be included in the manuscript file after the legends of regular figures. See detailed instructions regarding Expanded View files here: <https://www.embopress.org/page/journal/14602075/authorguide#expandedview>

- For the figures that you do NOT wish to display as Expanded View figures, they should be bundled together with their legends in a single PDF file called "Appendix", which should start with a short Table of Contents (including page numbers). Appendix figures should be referred to in the main text as: "Appendix Figure S1, Appendix Figure S2" etc. Please see detailed instructions here: <https://www.embopress.org/page/journal/14602075/authorguide#expandedview>

- A complete author checklist, which you can download from our author guidelines (<https://www.embopress.org/page/journal/14602075/authorguide>). Please note that the checklist will also be part of the Review Process File.

2. Please note that no statistics should be calculated and shown in Figures if $n=2$. Please also note that each p value should be reported as an exact value.

3. Before submitting your revision, primary datasets (and computer code, where appropriate) produced in this study need to be deposited in appropriate public databases (see <https://www.embopress.org/page/journal/14602075/authorguide#dataavailability>). In particular, all datasets such as DNA or RNA sequencing data, or mass spectrometry data generated in the study must be deposited in appropriate repositories. The accession numbers, database, and the specific URLs (links) should be listed in a formal "Data availability" section (placed after Methods), following the example below:

"The RNA-seq datasets produced in this study are available in the following database:
Gene Expression Omnibus GSE46843 (<https://www.ncbi.nlm.nih.gov/geo/query/acc.cgi?acc=GSE46843>)"

*** All links should resolve to a page where the data can be accessed. ***

*** Please remember to provide in the Data availability section of your revised manuscript reviewer passwords if the datasets are not yet public. ***

*** The Data Availability Section is restricted to new primary data that are part of this study. In case you have no data that require deposition in a public database, please state so instead of referring to the database: "Our study includes no data deposited in public repositories." under the heading "Data availability". ***

4. The materials and methods need to be described in the manuscript using our structured methods format, which is now required for all research articles. According to this format, the Methods section includes a single "Reagents and Tools Table" - listing key reagents, experimental models, software and relevant equipment including their sources and relevant identifiers - followed by a "Methods and Protocols" section describing the methods. Please download and fill our Reagents and Tools Table template (.docx), which you can find in our author guide: <https://www.embopress.org/page/journal/14602075/authorguide#structuredmethods>. When submitting your revised manuscript, please do not include the Reagents and Tools Table in the Methods section of the manuscript but instead upload it as a separate file choosing the file type "Reagent Table".

5. Please check that the title and the abstract of the manuscript are brief, yet explicit, even to non-specialists. The length of the title should not exceed 100 characters, and the abstract should be a single paragraph not exceeding 175 words.

6. Please also note our reference format: <https://www.embopress.org/page/journal/14602075/authorguide#referencesformat>.

8. Please remember: digital image enhancement is acceptable practice, as long as it accurately represents the original data and conforms to community standards. If a figure has been subjected to significant electronic manipulation, this must be noted in the

figure legend or in the "Materials and Methods" section. The editors reserve the right to request original versions of figures and the original images that were used to assemble the figure.

9. Our journal encourages inclusion of data citations in the reference list to directly cite datasets that were obtained from public databases. Data citations in the article text are distinct from normal bibliographical citations and should directly link to the database records from which the data can be accessed. In the main text, data citations are formatted as follows: "Data ref: Smith et al, 2001" or "Data ref: NCBI Sequence Read Archive PRJNA342805, 2017". In the Reference list, data citations must be labeled with "[DATASET]". A data reference must provide the database name, accession number/identifiers, and a resolvable link to the landing page from which the data can be accessed at the end of the reference. Further instructions are available at: <https://www.embopress.org/page/journal/14602075/authorguide#referencesformat>.

10. We request authors to consider both actual and perceived competing interests. Please review our policy (<https://www.embopress.org/page/journal/14602075/authorguide#conflictsofinterest>) and update your competing interests statement if necessary. Please name this section 'Disclosure and competing interests statement' and place it after the Acknowledgements section.

11. Please note that all corresponding authors are required to provide an ORCID ID upon submission of a revised manuscript (<https://orcid.org/>). Please find instructions on how to link your ORCID ID to your account in our manuscript tracking system in our Author guidelines (<https://www.embopress.org/page/journal/14602075/authorguide#authorshipguidelines>).

12. We use CRediT to specify the contributions of each author in the journal submission system. CRediT replaces the author contribution section, which should be removed from the manuscript. Please use the free text box to provide more detailed descriptions. See also guide to authors: <https://www.embopress.org/page/journal/14602075/authorguide#authorshipguidelines>.

14. We would also welcome the submission of cover suggestions or motifs to be used by our Graphics Illustrator in designing a cover.

15. Please use the link below to submit your revision:
<https://emboj.msubmit.net/cgi-bin/main.plex>

Referee #1:

Kläsener et al. propose that CD20 not only maintains the resting state of B cells but also orchestrates the microtubule/actin switch in activated B lymphocytes. The mechanism revealed herein demonstrates that CD20 functions as a "signal transducer," providing a theoretical basis for optimizing anti-CD20 therapies.

The following issues need to be addressed:

1. The study suggests that 14-3-3 acts as an adaptor protein linking CD20 and GEF-H1, but the molecular details of their interaction remain unclear. Are there other auxiliary proteins involved in this process?
2. The 14-3-3 family comprises multiple isoforms (e.g., 14-3-3 σ , ϵ , etc.) with functional isoform-specificity. Do different isoforms exhibit differences in their binding to CD20?
3. Is microtubule depolymerization a necessary prerequisite for CD20-mediated signal transduction?
4. Is page 34 a blank page with no content displayed?
5. The clarity of some figures is rather blurry, such as fig3A and fig3B. It is recommended to use an appropriate dpi for display?
6. Does the dissociation of the GEF-H1/CD20 complex upon anti-CD20 antibody treatment occur simultaneously with microtubule depolymerization, or is there a temporal sequence?
7. Different figures present P-values in different forms, such as Fig2F and Fig5B, fig7A and Fig7D. It is recommended to use a more uniform way to present statistical graphs.

Referee #2:

CD20 is not only an important functional receptor in B cells but also a target for immunotherapy in B cell malignancies and autoimmune diseases. CD20 is found in surface IgD nanoclusters in naïve B cells where it is thought to regulate the stability of

the BCR. mAbs against CD20 deplete B cells via ADCC but can also directly induce signaling and/or apoptosis in B cells, suggesting that CD20 is subject to distinct modes of regulation in naïve and activated B cells.

In this study, the authors investigate a novel mechanism by which CD20 exerts regulation in both naïve and activated B cells. Using human lymphoma cell lines and primary naïve human B cells, they demonstrate that CD20 is constitutively phosphorylated by PKC at specific residues in the cytosolic tails. This leads to its coupling with the microtubule network via interaction with 14-3-3 and RhoGEF-H1. Binding of Abs to CD20 causes dissolution of microtubules, switching its binding from RhoGEF-H1 to Rho-ROCK1, which ultimately induces F-actin generation.

Using a combination of gene expression, phosphorylation, immunoprecipitation and PLA, the findings from this study suggest a new paradigm in B cell biology that relies on phosphorylation-dependent switching of CD20 between microtubule and F-actin coupling in naïve vs. rituximab-treated cells. The conclusions are largely supported by experimental data. Co-localization analyses with PLA are a particularly strong aspect of the results. However, the following comments should be addressed to substantiate the conclusions and clarify the overall message:

1. The introduction can be focused more to better describe the rationale for this investigation.
2. In Fig 1 the authors show that PKC KO results in upregulation of CD69 expression, as well as that of other cell surface receptors. It is not clear if upregulation of CD69 and other receptors is simply due to greater trafficking/partitioning to the surface or true activation that is accompanied by phosphorylation of proximal signaling proteins and downstream mediators such as MAPK pathway. Are other markers of B cell activation (MHC II, CD44, CD80) also upregulated as a result of PKC ?
3. In Fig S2A and S2B gene expression pathway analysis is shown, and focuses on MT disassembly and cortical actin stability genes - what are some examples of these two sets of genes whose expression is altered PKC KO B cells? The legend for this figure has ambiguous phrasing.
4. In Fig 3F, the IP of CD20 with 14-3-3 is not clear and loading of CD20 is uneven across lanes. Similar concern about CD20 and GEF-H1 IP in Fig. 4D. Quantification of the IPs should be included.
5. In Fig S5, it is important to show the representative western blots of phosphorylation in addition to quantification of pGEF-H1, RhoA and Rock1.
6. It is not clear if Rock1 can directly phosphorylate CD20 and if this plays a role in mediating connection with BCR signaling machinery and the actin cytoskeleton.
7. It is mentioned that dissolution of the MT network and subsequent apoptosis of B cells upon CD20 crosslinking explains the therapeutic efficacy of agents such as rituximab. However, the functional consequence of the switch from MT to coupling with the actin cytoskeleton in the presence of rituximab is not especially meaningful, as the B cell is already undergoing apoptosis under these conditions and not likely to simultaneously undergo activation. Accordingly, the relevance of CD20 co-localization with the IgM BCR withing microvillar actin network in the context of anti-CD20 is also unclear. This model also does not explain how cognate Ag binding to the BCR causes CD20 co-localization.

Referee #3:

The manuscript by Kläsener et al, "Microtubule anchoring and coupling of CD20 to the RhoA/Rock1 pathway" presents interesting findings and sheds light on the cellular responses triggered by widely used anti-CD20 therapy. The model of microtubule-mediated coupling of different signalling pathways is novel and interesting. However, the manuscript seems somewhat unfinished and would benefit from a major revision.

In general, the flow and logic of the manuscript should be improved, together with robustness in terms of experimentation and statistical analyses. The manuscript appears somewhat jumpy, transitioning from one cell model to another without adequate explanations. The beginning of the manuscript seems like a collection of experiments without a clear reasoning.

Specific concerns:

Figure 1. A. The data with PKCd is presented superficially. The authors generate two clones of Ramos B cells lacking PKCd. It appears that only one of them is used for. However, there is a good chance for clonal effects when generating CRISPR-KO lines, which should be controlled.

B. The volcano plot shows two dots for CD20: S254. How is this possible? Also, the phosphoproteomics analysis should be illustrated more comprehensively. There are many other hits in this dataset. Fig. S1C shows more, but the majority of the data remains hidden. The full phosphoproteomics dataset, at least the sites with significant fold-change differences, should be included as part of the publication, in accordance with the principles of open science.

Fig S2. Very little information is given in the text or in the figure on this dataset. The data is mostly hidden. The figure is unclear and appears unfinished. It remains unclear on what basis are the MT and actin-related genes selected, and what are these genes. The context and meaning of this data seems unfinished in the text.

Figure 2E. The experimental setup nicely shows the specificity of the PLA assay. However, the statement of the localization to

the same nanoclusters is challenging as the domains are small and dynamic. There can also be proximal molecular localizations between different domains. The amount of PLA pairing components would inevitably affect the amount of signal obtained in this assay. How can the authors separate the effect of surface expression from being in the same membrane nanodomain? Additional supporting experimentation would be needed here.

On Row 152, the authors state that the CD32b-tr Ramos cells have more stable IgD clusters. It remains unclear how this stability has been measured and how do the stability features compare to PKCd-KO line.

Row 176. The authors write "double transfectants". It is not clear what the second transfected construct is. Also, for the PLA, Figure 3D-E, the text is unclear what constructs are actually expressed in the cells and what has been knocked out. The cell-based tools need to be clarified for all parts, and the motivation for the combinations should be justified. If understood correctly, CD32b-tr is also used for CD20 mutant constructs. How would it affect the CD20 interactions if CD32b is not overexpressed?

The flow cytometry and Western blot data mostly lack statistics and information on the number of experiments. Eg. Figure 4D-F are key findings for this manuscript and need to be appropriately replicated and statistically analysed.

Towards the end, the work completely shifts to use HD B cells. What experiments have been done with both cell models and how does the MT cytoskeleton treatment affect Ramos?

minor comments:

Row 90, " a member of the novel serine/threonine protein kinase C (PKC) family". The PKC family is well recognized and identified decades ago. Why "novel"?

figure S5. The corresponding example flow data should be included.

Referee #1:

Kläsener et al. propose that CD20 not only maintains the resting state of B cells but also orchestrates the microtubule/actin switch in activated B lymphocytes. The mechanism revealed herein demonstrates that CD20 functions as a "signal transducer," providing a theoretical basis for optimizing anti-CD20 therapies.

Thank you for your judgement that our study of the CD20 function provides a theoretical basis for optimizing anti-CD20 therapies.

The following issues need to be addressed:

1. The study suggests that 14-3-3 acts as an adaptor protein linking CD20 and GEF-H1, but the molecular details of their interaction remain unclear. Are there other auxiliary proteins involved in this process?

CD20 and GEF-H1, both carry an 14-3-3 binding sequence (RxxSxP) with the serine being constitutively phosphorylated in resting B lymphocytes. We thus think that the dimeric adaptor protein 14-3-3 can directly connect the two proteins without other auxiliary proteins.

2. The 14-3-3 family comprises multiple isoforms (e.g., 14-3-3 σ , ϵ , etc.) with functional isoform-specificity. Do different isoforms exhibit differences in their binding to CD20?

Ramos cells highly express the 14-3-3 isoforms beta (YHWAB), epsilon (E), eta (H), gamma (G), theta (Q), and zeta/delta (Z). Naïve HD B cells also express 14-3-3 sigma, though this isoform is expressed less well in Ramos cells. Using the PLA assay, we examined CD20:14-3-3 β , ϵ , and ζ binding in Ramos and naïve B cells. The latter cells also showed constitutive CD20:14-3-3 sigma interaction. As previously published, 14-3-3 binding to its target sequence is somewhat redundant, meaning several 14-3-3 isoforms can connect CD20 to GEF-H1. Furthermore, these 14-3-3 isoforms frequently exist as heterodimers^{1,2}.

3. Is microtubule depolymerization a necessary prerequisite for CD20-mediated signal transduction?

We believe this to be the case, which is one of the major findings of our study. Figure 7 shows that pretreating HD B cells with docetaxel, which prevents microtubule depolymerization, blocks RTX-induced CD20 signaling, including Rock1 phosphorylation and IgM-BCR/CD19 conjugation. Conversely, these CD20 signaling events are induced by the microtubule depolymerization agent nocodazole in the absence of RTX.

4. Is page 34 a blank page with no content displayed?

Thank you for pointing out the formatting error in our manuscript, which we have corrected in the new version.

5. The clarity of some figures is rather blurry, such as fig3A and fig3B. It is recommended to use an appropriate dpi for display?

We have now used a higher-resolution version of the drawings in Figures 3A and 3B, and we have adjusted the resolution of all figures accordingly.

6. Does the dissociation of the GEF-H1/CD20 complex upon anti-CD20 antibody treatment occur simultaneously with microtubule depolymerization, or is there a temporal sequence?

As shown in figures 6C and 5D, a 5 min exposure of HD B cells to RTX results in dissociation of the GEF-H1/CD20 complex. In figure S7B, we demonstrate that a 5 min exposure of HD B cells to RTX alters the microtubule network stained with anti-tubulin antibody. Therefore, based on the data shown in Figure 7, we conclude that these two events occur simultaneously and are tightly coupled.

7. Different figures present P-values in different forms, such as Fig2F and Fig5B, fig7A and Fig7D. It is recommended to use a more uniform way to present statistical graphs.

Thank you for pointing this out. We have replaced all asterisks with numeric values (e.g. **** with 0.001) and these are now used in all the figures.

Referee #2

CD20 is not only an important functional receptor in B cells but also a target for immunotherapy in B cell malignancies and autoimmune diseases. CD20 is found in surface IgD nanoclusters in naïve B cells where it is thought to regulate the stability of the BCR. mAbs against CD20 deplete B cells via ADCC but can also directly induce signaling and/or apoptosis in B cells, suggesting that CD20 is subject to distinct modes of regulation in naïve and activated B cells.

In this study, the authors investigate a novel mechanism by which CD20 exerts regulation in both naïve and activated B cells. Using human lymphoma cell lines and primary naïve human B cells, they demonstrate that CD20 is constitutively phosphorylated by PKC δ at specific residues in the cytosolic tails. This leads to its coupling with the microtubule network via interaction with14-3-3 and RhoGEF-H1. Binding of Abs to CD20 causes dissolution of microtubules, switching its binding from RhoGEF-H1 to Rho-ROCK1, which ultimately induces F-actin generation.

Using a combination of gene expression, phosphorylation, immunoprecipitation and PLA, the findings from this study suggest a new paradigm in B cell biology that relies on phosphorylation-dependent switching of CD20 between microtubule and F-actin coupling in naïve vs. rituximab-treated cells. The conclusions are largely supported by experimental data. Co-localization analyses with PLA are a particularly strong aspect of the results. However, the following comments should be addressed to substantiate the conclusions and clarify the overall message:

Thank you for your judgement that our study suggests a new paradigm in B cell biology.

1. The introduction can be focused more to better describe the rationale for this investigation.

In the introduction to the new version of our manuscript, we now describe the rationale for our study more clearly. Namely, we aim to explore how CD20 supports the resting state of B cells and how the two other known B-cell gatekeepers, CD32b and PKC- δ , cooperate with CD20 to fulfill this task.

2. In Fig 1 the authors show that PKC δ KO results in upregulation of CD69 expression, as well as that of other cell surface receptors. It is not clear if upregulation of CD69 and other receptors is simply due to greater trafficking/partitioning to the surface or true activation that is accompanied by phosphorylation of proximal signaling proteins and downstream mediators such as MAPK pathway. Are other markers of B cell activation (**MHC II, CD44, CD80**) also upregulated as a result of PKC δ ?

Ramos B cells do not express CD44, but our new figure 2 shows that PKC δ knockout (KO) Ramos cells also upregulate MHCII and CD86. Our manuscript focuses on the constitutive

phosphorylation of CD20 tail serine residues by PKC δ , rather than on the study of PKC δ KO B cells, which have been previously described^{3,4}.

3. In Fig S2A and S2B gene expression pathway analysis is shown, and focuses on MT disassembly and cortical actin stability genes - what are some examples of these two sets of genes whose expression is altered PKC δ KO B cells? The legend for this figure has ambiguous phrasing.

We have now provided a more detailed description of this figure and named some of the key genes whose expression is altered in PKC δ KO B cells as a table in figure S2.

4. In Fig 3F, the IP of CD20 with 14-3-3 is not clear and loading of CD20 is uneven across lanes. Similar concern about CD20 and GEF-H1 IP in Fig. 4D. Quantification of the IPs should be included.

In figure 3F and figure 4D we now show a quantification and the number of the IP experiments (n=3).

5. In Fig S5, it is important to show the representative western blots of phosphorylation in addition to quantification of pGEF-H1, RhoA and Rock1.

We think this is comment resulted from a misunderstanding of our data. The graphs show in figure S5 are not derived from western blot experiments but rather from intracellular phospho-flow cytometry experiment (n=6)

6. It is not clear if Rock1 can directly phosphorylate CD20 and if this plays a role in mediating connection with BCR signaling machinery and the actin cytoskeleton.

We do not think that ROCK1 phosphorylates the tail of CD20 as they are already phosphorylated by PKC δ as we shown in the new figure 2. Furthermore, CD20 has never been described as a ROCK1 substrate. A well-known substrate of ROCK1 we mention in our manuscript is the myosin light chain II. Apart from this, Rock1 has several substrates that interact with and/or promote the formation of actin filaments. In our 1-PLA study, we detected a close proximity between CD20 and Rock1 in RTX treated Ramos (figure 5 F, H) or HD B cells (figure 6 G, H).

In our 1-PLA study, we detected close proximity between CD20 and ROCK1 in RTX-treated Ramos cells (figure 5, F and H) and HD B cells (figure 6, G and H). This increased proximity could result in several interactions, including the recruitment of ROCK1 to the inner leaflet of the plasma membrane via binding to RhoA-GTP and/or binding of the 14-3-3 adaptor to the C-terminus of ROCK1, which targets an RxxS1333 motif (with S1333 being a known autophosphorylation site).

7. It is mentioned that dissolution of the MT network and subsequent apoptosis of B cells upon CD20 crosslinking explains the therapeutic efficacy of agents such as rituximab. However, the functional consequence of the switch from MT to coupling with the actin cytoskeleton in the presence of rituximab is not especially meaningful, as the B cell is already undergoing apoptosis under these conditions and not likely to simultaneously undergo activation. Accordingly, the relevance of CD20 co-localization with the IgM BCR withing microvillar actin network in the context of anti-CD20 is also unclear. This model also does not explain how cognate Ag binding to the BCR causes CD20 co-localization

We and others do not find that the RTX treatment result in the full activation of the apoptosis program^{5,6} but in an increase Caspase 3 activity (data not shown). We think that the disassembly of the microtubule network renders the B cells more prone to apoptosis in context with other interventions such as ADCC, following an attack of RTX-bound B cells by NK cells.

In many previous experiments, we have demonstrated the colocalization of CD20 and CD19 with the (actin associated) IgM-BCR in activated B cells, and the association of RTX-bound CD20 with actin-based microvilli has recently been clearly shown⁷. Currently, we do not know the relevance of CD20 co-localization with the IgM BCR within the microvillar actin network, but we think that CD20-promoted transport of the RhoA-GTP/Rock1 complex to the IgM-BCR could stabilize the microvillar actin network. We think that the connection of RTX-bound CD20 to the actin cytoskeleton is also therapeutical relevant and our insights could help to stabilize surface CD20 expression and to prevent its rapid internalization.

Referee #3:

The manuscript by Kläsener et al, "Microtubule anchoring and coupling of CD20 to the RhoA/Rock1 pathway" presents interesting findings and sheds light on the cellular responses triggered by widely used anti-CD20 therapy. The model of microtubule-mediated coupling of different signalling pathways is novel and interesting. However, the manuscript seems somewhat unfinished and would benefit from a major revision.

Thank you for your judgement that our finding of the microtubule-mediated coupling of CD20 to different signalling pathways as novel and interesting.

In general, the flow and logic of the manuscript should be improved, together with robustness in terms of experimentation and statistical analyses. The manuscript appears somewhat jumpy, transitioning from one cell model to another without adequate explanations. The beginning of the manuscript seems like a collection of experiments without a clear reasoning.

The path to a scientific discovery is sometimes a winding road but in the new version of our manuscript, we changed the flow and provide a clear reasoning for our study namely to explore the cellular mechanism of the gatekeeper function of CD20 and its interactions with two other known gatekeepers of resting B cells. We think that our study is important because, although there are hundreds of publications describing signaling mechanisms in activated lymphocytes, only a few study the resting state, which provides the base line for any activation process. In our study we only use two types of cells namely healthy donor B cells and Ramos cells, which can be altered using established engineering tools, such as CRISPR/Cas9 and vector transfections.

Specific concerns:

Figure 1. A. The data with PKC δ is presented superficially. The authors generate two clones of Ramos B cells lacking PKC δ . It appears that only one of them is used for. However, there is a good chance for clonal effects when generating CRISPR-KO lines, which should be controlled.

It is true that we studied only one of the two Ramos PKC δ -KO line by the mass spectrometry. However, the goal of our project was not to study the cellular function of PKC δ , which would certainly require more mutant cell lines, but that of CD20. The MS/MS data we obtained allowed us to identify the serine residues in the tails of CD20 that are phosphorylated by PKC δ . We then proved the importance of these serine residues our extensive mutational study, which is an important part of our manuscript.

B. The volcano plot shows two dots for CD20: S254. How is this possible? Also, the phosphoproteomics analysis should be illustrated more comprehensively. There are many other hits in this dataset. Fig. S1C shows more, but the majority of the data remains hidden. The full phosphoproteomics dataset, at least the sites with significant fold-change differences,

should be included as part of the publication, in accordance with the principles of open science.

The two dots of pS254 in the C-terminal tail of CD20 arise because the generated peptide contains two serine residues S253/S254 that can be either alone or doubly phosphorylated by PKC δ . To clarify this point for the reader, we have now labeled each data point in the new Figure 2 with its corresponding multiplicity.

We have no intention to withhold our mass spectrometry data. The complete raw dataset, including all MaxQuant search results, have been made available via the PRIDE repository (PXD063667) prior to submission. All phospho-proteomics information is included in the Phospho (STY)Sites.txt file, which can be opened using conventional spreadsheet software. For convenience, we now also provide a supplementary table summarizing all significant phosphosites.

Fig S2. Very little information is given in the text or in the figure on this dataset. The data is mostly hidden. The figure is unclear and appears unfinished. It remains unclear on what basis are the MT and actin-related genes selected, and what are these genes. The context and meaning of this data seems unfinished in the text.

Thank you for pointing this out. We have now provided a more detailed description of this figure and identified some of the key genes whose expression is altered in PKC δ KO B cells presented in a table in Supplementary figure S2.

Figure 2E. The experimental setup nicely shows the specificity of the PLA assay. However, the statement of the localization to the same nanoclusters is challenging as the domains are small and dynamic. There can also be proximal molecular localizations between different domains. The amount of PLA pairing components would inevitably affect the amount of signal obtained in this assay. How can the authors separate the effect of surface expression from being in the same membrane nanodomain? Additional supporting experimentation would be needed here.

It is true that our PLA only monitors the proximity of the proteins under study on the surface of fixed B cells and we cannot make any judgement on the size and dynamics of the nanocluster. However, super resolution studies of receptor proximity on living cells do not easily reach the nanoscale resolution of our PLA study. According to referee 2, our "co-localization analyses with PLA are a particularly strong aspect of the results". These PLA studies allow us to obtain knowledge of the relative proximity of a protein **especially on the cell surface**. To demonstrate this, we present now the new figure1, which shows not only the IgD:CD32b, but also the IgM:CD32b PLA data from the CD32b-tr Ramos B cells. These data indicate that CD32b, an inhibitory receptor, is part of the IgD-BCR rather than the IgM-BCR nanocluster.

On Row 152, the authors state that the CD32b-tr Ramos cells have more stable IgD clusters. It remains unclear how this stability has been measured and how do the stability features compare to PKC δ -KO line.

Referee 3 is right that we cannot make any judgement on the stability of the IgD clusters based on our data. We can only state that proteins are more abundant on the surface of CD32b-tr than on WT Ramos B cells when they are localized in close proximity to the IgD-BCR. We now avoid the word "stability" in the text of our manuscript.

Row 176. The authors write "double transfectants". It is not clear what the second transfected construct is. Also, for the PLA, Figure 3D-E, the text is unclear what constructs are actually expressed in the cells and what has been knocked out. The cell-based tools need to be clarified for all parts, and the motivation for the combinations should be justified. If understood

correctly, CD32b-tr is also used for CD20 mutant constructs. How would it affect the CD20 interactions if CD32b is not overexpressed?

We thought that in row 173 to 175 we clearly explained the nature of the CD20 + CD32-tr double-transfectants “double-transfectants expressed CD20 and CD32b at levels similar to the CD32b-tr Ramos line”. In the new version of our manuscript, we better explain why we used the CD32b-transfected Ramos line in our study as they are more similar to healthy donor B cells in their CD32b and CD20 expression (see new figure 1A). We also performed experiments on single CD20 mutant Ramos cells with comparable results; however, due to the lower expression of the IgD-BCR nanocluster components, the data had reduced amplitude.

The flow cytometry and Western blot data mostly lack statistics and information on the number of experiments. Eg. Figure 4D-F are key findings for this manuscript and need to be appropriately replicated and statistically analysed.

In figure 4D-F we now show a quantification of the IP experiments (n=3).

Towards the end, the work completely shifts to use HD B cells. What experiments have been done with both cell models and how does the MT cytoskeleton treatment affect Ramos?

In the figure 5 and figure 6 we show the PLA studies done with Ramos and, for comparison, with HD B cells, respectively.

minor comments:

Row 90, " a member of the novel serine/threonine protein kinase C (PKC) family". The PKC family is well recognized and identified decades ago. Why "novel"?

The name “novel PKC (nPKC)” is used in the literature when referring to the PKC family members PKC δ (delta), PKC ϵ (epsilon), PKC η (eta) and PKC θ (theta). Novel PKCs require DAG for their activation but not Ca²⁺⁸.

figure S5. The corresponding example flow data should be included.

figure S5 is now converted to figure S4. Here, we now show examples of the quantified phosphor-flow cytometry data.

Further References:

1. Obsilova, V. & Obsil, T. Structural insights into the functional roles of 14-3-3 proteins. *Frontiers in Molecular Biosciences* vol. 9 (2022).
2. Xiao, B., Smerdon, S. J., Jones, D. H., Dodson, G. G., Soneji, Y., Aitken, A., & Gamblin, S. J. (1995). Structure of a 14-3-3 Protein and Implications for Coordination of Multiple Signalling Pathways. *Nature*, 376(6536), 188–191.
3. Sun Kuehn, H. et al. Loss-of-function of the protein kinase C d (PKCd) causes a B-cell lymphoproliferative syndrome in humans. *Blood* 121, 3117–3125 (2013).
4. Miyamoto. Increased proliferation of B cells and auto-immunity in mice lacking protein kinase Cd. *Nature* 865 (2002).
5. Rezvani, A. R. & Maloney, D. G. Rituximab resistance. *Best Practice and Research: Clinical Haematology* vol. 24 203–216 (2011).

6. Smith, M. R. Rituximab (monoclonal anti-CD20 antibody): Mechanisms of action and resistance. *Oncogene* 22, 7359–7368 (2003).
7. Kozlova, V. et al. CD20 is dispensable for B-cell receptor signaling but is required for proper actin polymerization, adhesion and migration of malignant B cells. *PLoS One* 15, (2020).
8. Barbazuk, S. M. & Gold, M. R. Protein Kinase C-Delta Is a Target of B-Cell Antigen Receptor Signaling. *Immunology Letters* vol. 69 www.elsevier.com/locate/ (1999).

Dear Michael, dear Kathrin,

Thank you again for the submission of your revised manuscript (EMBOJ-2025-122022R) to The EMBO Journal for our consideration, and for your patience during re-review. Your revised manuscript has now been seen by the three original referees, who had previously reviewed the first version of the manuscript, and we have received their comments, which are appended below.

I am very pleased to say that, as you will see, all three referees are satisfied with the revision, recognize that the initially raised criticisms and concerns have been adequately addressed in an improved version of the manuscript, and now endorse the publication of this manuscript without any other major concerns.

There are only a few remaining points (from referees #2 and #3) for minor corrections in the Figures and their legends, which I kindly request you address in a final version of your manuscript. Please include in your resubmission a point-by-point response detailing any changes to the manuscript.

From the editorial side, there are also a number of changes and corrections we need you to make in the final version of your manuscript, before we can process it further for publication in The EMBO Journal:

- Thank you for providing reviewer access to your deposited mass spectrometry and RNA-sequencing datasets. Confidential reviewer access codes/tokens can now be removed from the Data Availability statement; please make sure that the datasets will be publicly available at the time of publication, and update this section with the databases, accession IDs, and permanent URLs to the datasets. Please also move the GEO accession ID and link from paragraph "GEO data" to the same Data availability section.

- The author contributions statement should be removed from the manuscript file. Instead, we use CRediT to specify the contributions of each author in the journal submission system. Please feel free to use the free text box to provide more detailed descriptions during submission. See also our guide to authors for more information: <https://link.springer.com/partners/embo-press/editorial-policies#Authorship>.

- Please change heading "Disclosure and competing interest statement" to "Discosure and competing interests statement".

- We noticed that one of your co-authors has an affiliation with a pharmaceutical company (affiliation #7). According to our journal's policy, employment in a a biotechnology/pharmaceutical company must be stated in the "Disclosure and competing interests statement".

- Please note that the References format needs to be updated to the style used by EMBO Press. In particular, the list must be alphabetical (not numerical), and "et al." must be used after the names of the first 10 co-authors of each citation. In addition, DOIs should only be provided for preprints and datasets that have not been published yet. Please check our author guidelines for more information on our References format: <https://link.springer.com/journal/44318/submission-guidelines#cms-Reference-guidelines>.

- Please correct the journals' name in your Author Checklist to "The EMBO Journal" (in the general information table at the top of the checklist). In addition, please use the last column of the checklist ("In which section is the information available?") only for listing the manuscript sections where the relevant information can be found; the information/explanations/descriptions themselves must be included in the manuscript, not in the checklist. If the provided DOI in the "Design" section refers to the preprint of this manuscript, it does not need to be listed under the pre-registered study protocols section.

- Please make sure that the funding information provided in the Acknowledgements section of the manuscript matches the information entered in our online manuscript handling system. Currently, information on "WA 1597/3-1" is missing in the online system. In addition, the funders listed in the Comments box need to be removed so that each one is provided separately using the "More Funders" option. Funding information in the manuscript needs to be provided in the Acknowledgments section, no separate section is needed.

- It appears that you prefer the information provided in "Table S2" to be a part of Appendix Figure S2 (as a third panel). This is totally fine, but in that case it should neither be provided as a separate file nor called out as a separate item/table - it would only be called out as "Appendix Figure S2C", throughout the manuscript and the Appendix file.

- Please rename the separately uploaded Table 1 to "Table EV1", and the Excel file titled "PKCdelta KO phospho proteome" to "Dataset EV1" (with its legend in a separate tab in the same Excel file). Please make sure that callouts are updated accordingly through the manucript and related files.

- The Appendix file needs to be uploaded in PDF format; its title (first) page should contain the heading "Appendix for:", followed

by the manuscript's title and a Table of Contents including page numbers for all listed items. "Table S2C" should be removed from the list (please also see more detailed comments on this table above). The correct nomenclature for the Appendix items is "Appendix Figure S#" or "Appendix Table S#", and this must be updated accordingly throughout the main manuscript and all related files.

- Please note that EMBO press papers are accompanied online by:

A) a short (2 sentences) summary of the findings and their significance,

B) 2-5 short bullet points highlighting the key results, and

C) a synopsis image in .jpg or .png format that is exactly 550 pixels wide and 300-600 pixels high (the height is variable). Please note that all text needs to be legible at the final size.

Please upload this information along with your revised manuscript (the text for A and B should be provided in a separate Word file).

- The "Structured Methods" text should be provided as "Methods and Protocols" under "Methods" in the main manuscript file, not in the separate "Reagents and Tools Table".

- Thank you for uploading the Source Data for your Figures and the completed Source Data checklist. We kindly request you reorganize these Data and upload the Data for each main Figure separately; for example all Source Data for Figure 1 panels should be uploaded in a separate zip folder named "Figure 1 Source Data.zip". Any Source Data for EV and/or Appendix Figures can be uploaded in one zip folder names "EV Source Data.zip".

- Please change heading "Materials & Methods" to "Methods".

- Our data editors have checked your Figures and their legends, and raised the following queries. Please address all points below completely in your revised manuscript (all changes should be highlighted or "tracked"):

- Please note that the legends for Figures 5, 6 are not provided in the sequential manner. This needs to be rectified.

- Please provide the exact p-values in the legends of Figures 1E, 3E, 4C, 5B, D, F, H; 6B, D, F, H; 7E, F.

- Please indicate the statistical test used for data analysis in the legends of Figures 3F, 4D.

- Please note that information related to "n" is missing in the legends of Figures 2B, 4C, 5B, D, F, H; 6B, D, F, H; 7A, B, E, F.

- Although "n" is provided, please describe the nature of entity for "n" in the legends of Figures 2A, 3E, 4D.

- Please note that the error bars must be defined in the legends of Figures 2A, 3F, 4C, D, 6B, D, F, H; 7A, B, E, F.

- The manuscript sections need to be named and ordered as follows: Title page - Abstract - Introduction - Results - Discussion - Methods - Data Availability - Acknowledgements - Disclosure and Competing Interests Statement - References - Figure Legends - main Tables (if there are any) - Expanded View Figure Legends.

- Please also note that as part of the EMBO Press transparent editorial process, The EMBO Journal publishes online a Peer Review File along with each accepted manuscript. This File will be published in conjunction with your paper and will include the referee reports, your point-by-point responses and all pertinent correspondence relating to the manuscript. Your Author's Checklist will also be published at the end of the Peer Review File. Please let us know in case you want to remove any data or figures from your point-by-point responses before they are published as part of the Peer Review File. Retaining unpublished data in the Peer Review File means that these count as published and that the Peer Review File would need to be referenced in future publications. Please let the editorial office know in case you want to remove any data from this file (contact@embojournal.org).

We look forward to seeing a final version of your manuscript as soon as possible. Please let us know if you have any questions and use this link to submit your revision: <https://emboj.msubmit.net/cgi-bin/main.plex>.

Best regards,

Ioannis

Referee #1:

The authors have fully addressed all the previous review comments with appropriate revisions.

Referee #2:

Using elegant imaging and biochemical methods, the authors of this manuscript identify a novel mechanism by which CD20 regulates naïve and activated B cell activation. They show that CD20 is constitutively phosphorylated by PKC at specific residues in the cytosolic tails, which enables its coupling with the microtubule network via interaction with 14-3-3 and RhoGEF-H1. Binding of Abs to CD20 causes dissolution of microtubules, switching its binding from RhoGEF-H1 to Rho-ROCK1, which ultimately induces F-actin generation. Thus, the demonstration that CD20 serves as an important regulatory switch in the naïve to active B cell transition should appeal to the broad readership of The EMBO Journal.

The revisions adequately address my concerns, except for a minor typographical change - In Fig. S3 legend parts A and B should be correctly labeled.

Referee #3:

I thank the authors for revising their manuscript. The manuscript now flows better, and the data is presented more clearly. I do not have further major concerns.

As minor comments, I still feel that the figures have a slightly unfinished touch and could be made sleeker by optimizing the different font sizes and graph styles.

Figure 1A has Cd32b-KO included in the annotation, however not shown in the panel.

Figure 2 is missing from the merged file, while Figure 3 is duplicated.

Referee #1:

The authors have fully addressed all the previous review comments with appropriate revisions.

Thank you!

Referee #2:

Using elegant imaging and biochemical methods, the authors of this manuscript identify a novel mechanism by which CD20 regulates naïve and activated B cell activation. They show that CD20 is constitutively phosphorylated by PKC δ at specific residues in the cytosolic tails, which enables its coupling with the microtubule network via interaction with 14-3-3 and RhoGEF-H1. Binding of Abs to CD20 causes dissolution of microtubules, switching its binding from RhoGEF-H1 to Rho-ROCK1, which ultimately induces F-actin generation. Thus, the demonstration that CD20 serves as an important regulatory switch in the naïve to active B cell transition should appeal to the broad readership of The EMBO Journal.

The revisions adequately address my concerns, except for a minor typographical change - In Fig. S3 legend parts A and B should be correctly labeled.

Thank you for pointing this out. The figure was changed in the meantime. The labeling was corrected.

Referee #3:

I thank the authors for revising their manuscript. The manuscript now flows better, and the data is presented more clearly. I do not have further major concerns.

Thank you!

As minor comments, I still feel that the figures have a slightly unfinished touch and could be made sleeker by optimizing the different font sizes and graph styles.

We now revised the figures, striving for uniformity in font and size.

Figure 1A has Cd32b-KO included in the annotation, however not shown in the panel.

Thank you for pointing this out. The labeling in 1A included the cell stainings used in figures 1B and 1C. To avoid any misunderstanding, we separated the labeling according to the different sections B and C.

Figure 2 is missing from the merged file, while Figure 3 is duplicated.

We have revised all figures and uploaded the new versions to The EMBO Journal's online platform.

Dear Michael, dear Kathrin,

Congratulations on an excellent manuscript! I am very pleased to inform you that it has been accepted for publication in The EMBO Journal. Thank you for comprehensively addressing the initially raised referee concerns and our editorial requests for changes and corrections.

You may qualify for financial assistance for your publication charges - either via a Springer Nature fully open access agreement or an EMBO initiative. Check your eligibility: <https://link.springer.com/journal/44318/how-to-publish-with-us>

If you have any questions, please do not hesitate to contact the Editorial Office. Thank you for your contribution to The EMBO Journal. Working with you has been a pleasure.

Best regards,

Ioannis

Please note that it is The EMBO Journal policy for the transcript of the editorial process (containing referee reports and your response letters) to be published as an online supplement to each paper. If you should prefer removal of any referee-only figures included in the point-by-point response(s), e.g. because they may still be used for future publication or because they have been reproduced from published work by others, please do let us know immediately via response email. More information is available here: <https://link.springer.com/partners/embo-press/editorial-policies#Peer%20review>
